# EXPLORING THE LINK BETWEEN OUT-OF-DISTRIBUTION DETECTION AND CONFORMAL PREDICTION WITH ILLUSTRATIONS OF ITS BENEFITS

## ABSTRACT

Research on Out-Of-Distribution (OOD) detection focuses mainly on building scores that efficiently distinguish OOD data from In Distribution (ID) data. On the other hand, Conformal Prediction (CP) uses non-conformity scores to construct prediction sets with probabilistic coverage guarantees. In other words, the former designs scores, while the latter designs probabilistic guarantees based on scores. Therefore, we claim that these two fields might be naturally intertwined. This work advocates for cross-fertilization between OOD and CP by formalizing their link and emphasizing two benefits of using them jointly. First, we show that in standard OOD benchmark settings, evaluation metrics can be overly optimistic due to the test dataset's finite sample size. Based on the work of Bates et al. (2022), we define new *conformal AUROC* and *conformal FPR@TPR95* metrics, which are corrections that provide probabilistic conservativeness guarantees on the variability of these metrics. We show the effect of these corrections on two reference OOD and anomaly detection benchmarks, OpenOOD Yang et al. (2022) and ADBench Han et al. (2022). Second, we explore using OOD scores as non-conformity scores and show that they can improve the efficiency of the prediction sets obtained with CP.

## 1 INTRODUCTION

Even though current Machine Learning (ML) and Deep Learning (DL) models are able to perform several complex tasks that previously only human beings could, we are still a step away from their widespread adoption in safety-critical applications. Indeed, it is difficult to certify an ML component, mainly due to the poor control of the circumstances that may provoke such a ML component to fail. Out-of-Distribution (OOD) detection tries to tackle this problem by identifying data that differs significantly from the data used to train the model at runtime. Besides being recognized as an essential step in the certification of ML systems by multiple certification authorities (see, e.g., Sections 5.3 and 8.4 of Balduzzi et al. (2021) or Section 5.1 of EASA & Daedalean (2024)), OOD detection is a very active branch in machine learning research.

Current OOD detection strategies rely on constructing an OOD score $s$, a function that assigns a scalar to each input example. This score discriminates between in-distribution (ID) data and OOD data by assigning lower scores to the former and higher scores to the latter.

When OOD detection is used in a machine learning pipeline to identify examples that differ from the data the model has been trained on, there is a natural qualitative interpretation of OOD detection in terms of model uncertainty. For instance, an example with a low OOD score should be one for which the model can predict with low uncertainty, while an example with a high OOD score should be linked to a highly uncertain prediction.

Conformal Prediction (CP) is a family of post-hoc methods for Uncertainty Quantification and Uncertainty Representation Caprio et al. (2024), that work as wrappers over machine learning models, transforming point predictions into prediction sets with rigorous probabilistic guarantees based on so-called nonconformity scores. The user pre-specifies a risk level $\alpha$, and the constructed prediction set is guaranteed to contain the ground truth value with a probability of at least $1 - \alpha$. Since CP is a way of providing rigorous uncertainty quantification guarantees built upon scores, it is natural to apply it to the scores used in OOD detection. **The main purpose of our work is to dig into the**

**Conformal Prediction interpretation of OOD detection scores and show some of its advantages for both Conformal Prediction and OOD detection.**

To that end, we first follow the work of Bates et al. (2022) on outlier detection and apply their ideas to OOD detection. Bates et al. (2022) cast the OOD detection problem into the statistical framework of hypothesis testing. They show that the p-values, built with a calibration dataset, are provably marginally valid but depend on the choice of the calibration dataset, and so is the False Positive Rate (FPR) derived from these p-values. One of the main contributions of our work is to explore the consequences of this effect for OOD detection and to propose alternative *conformal AUROC* and *conformal FPR@TPR95* metrics.

The relevance of the new metrics we propose is best appreciated in the context of safety-critical applications, or in an eventual certification process of an OOD detection component. The true AUROC or FPR metrics are inaccessible for a given OOD score, and we can only provide an approximation obtained from a finite dataset. However, this can introduce fluctuations in our approximation, thus overestimating or underestimating the true metrics. In a certification process, we are mainly interested in guaranteeing that our estimations are conservative with high probability, at the expense of losing some approximation precision EASA (2023), which is precisely what Conformal AUROC and Conformal FPR do. We show the effect of these new metrics on two large reference benchmarks, the OOD benchmark OpenOOD Yang et al. (2022), and the anomaly detection benchmark Han et al. (2022).

Second, we show that not only can CP contribute to OOD detection, but research in OOD detection can also help CP. Indeed, CP has traditionally focused on constructing prediction sets from nonconformity scores. Still, the scores used are usually simple functions of the softmax scores for classification tasks or classical distances in Euclidean space for regression tasks. Here, we draw inspiration from the OOD detection literature to build more involved nonconformity scores and compare their performance to the traditional nonconformity scores of CP. For the task of classification, we build prediction sets based on multiple different OOD scores and find that some of them, notably Mahalanobis Leys et al. (2018) or KNN Sun et al. (2022), are good candidates as nonconformity scores.

Ultimately, one of the key messages of this work is that since OOD is concerned with designing scores and conformal prediction with interpreting these scores, the two fields may be inherently intertwined. **Highlighting this relationship might offer significant potential for cross-fertilization.**

Our contributions can be summarized as follows:

- We cast the OOD detection problem into the framework of statistical hypothesis testing and apply the ideas of Bates et al. (2022) to correct OOD scores and propose new conformal AUROC and conformal FPR@TPR95 metrics, which are provably conservative with high probability.
- We show the effect of conformal AUROC and conformal FPR in the reference benchmarks OpenOOD Yang et al. (2022) and ADBench Han et al. (2022).
- We build new nonconformity scores for CP based on OOD and perform a comparison between the scores. We find that the Mahalanobis score outperforms the classical CP score.
- We point out that OOD and CP are two domains that have much to contribute to each other and advocate for further research exploring this link.

## 2 BACKGROUND

**Out-of-Distribution Detection** Given $n$ examples, $\{\boldsymbol{x}_1, \ldots, \boldsymbol{x}_n\}$ sampled from a probability distribution $\mathcal{P}_{id}$ on a space $\mathcal{X}$, and a new data point $\boldsymbol{x}_{n+1}$, the task of Out-of-Distribution (OOD) detection consists in assessing if $\boldsymbol{x}_{n+1}$ was sampled from $\mathcal{P}_{id}$ - in which case it is considered In-Distribution (ID) - or not - thus considered OOD.

The most common procedure for OOD detection is to construct a score $s : \mathcal{X} \to \mathbb{R}$ and a threshold $\tau$ such that:

$$\begin{cases} \boldsymbol{x}_{n+1} \text{ is declared OOD if } s(\boldsymbol{x}_{n+1}) > \tau \\ \boldsymbol{x}_{n+1} \text{ is declared ID if } s(\boldsymbol{x}_{n+1}) \leq \tau \end{cases} \tag{1}$$

We call $s$ an OOD score.

**Task-based OOD**  This is the most common approach in the literature regarding OOD detection for neural networks. It also encompasses Open-Set Recognition. Let's consider that $\boldsymbol{x}_i$ can be assigned a label $y_i$ so that we can construct a dataset $\{(\boldsymbol{x}_1, y_1), \ldots, (\boldsymbol{x}_n, y_n)\}$ defining some supervised deep learning task. In that case, $\mathcal{P}_{id} := \mathcal{P}_{train}$. Task-based OOD uses representations built by the neural network $f$ throughout its training to design $s$. Many sophisticated methods follow this approach Yang et al. (2021). A simple example is to take the negative maximum of the output of $f$ (after the softmax) Hendrycks & Gimpel (2018) as an OOD score $(s(\boldsymbol{x}_{n+1}) = -\max(f(\boldsymbol{x}_{n+1})))$ where $\max(\boldsymbol{x})$ is the highest component of the vector $\boldsymbol{x}$. Another simple idea is to find the distance to the nearest neighbor in some intermediate layer of $f$ Sun et al. (2022).

**Task-agnostic OOD**  This approach encompasses One-Class Classification and Anomaly/Outlier Detection. Let's consider a dataset $\{\boldsymbol{x}_1, \ldots, \boldsymbol{x}_n\}$ in a fully unsupervised way. There is no notion of labels, so we have to approximate $\mathcal{P}_{id}$ somehow or some related quantities from scratch. Examples are GANs or VAEs with $s$ defined as reconstruction error. See Yang et al. (2021) for a thorough review.

**Conformal Prediction**  Few Machine Learning and Deep Learning models provide a notion of uncertainty related to their predictions. Even the models trained for classification tasks providing softmax outputs, which can be interpreted as the probabilities for the input belonging to the different classes, are usually ill-calibrated and overconfident, making the softmax output an incorrect proxy of the true uncertainty of the prediction. Pearce et al. (2021). Conformal Prediction (CP) Vovk et al. (2005); Angelopoulos & Bates (2022) is a series of post-processing uncertainty quantification techniques that are model-agnostic and provide finite-sample guarantees on the model predictions. One of the simplest CP techniques, the split CP, works as a wrapper on a trained model $f$. It requires a calibration dataset $\{(\boldsymbol{x}_{n+1}, y_{n+1}), \ldots, (\boldsymbol{x}_{n+n_{\mathrm{cal}}}, y_{n+n_{\mathrm{cal}}})\}$ independent of the training data, and a risk (or error rate) $\alpha$ that the user can tolerate. Based on so-called nonconformity scores computed on the calibration dataset, it builds a prediction set $C_\alpha(\boldsymbol{x}_{n+n_{\mathrm{cal}}+1})$ for a new test sample $\boldsymbol{x}_{n+n_{\mathrm{cal}}+1}$ with the following finite sample guarantee

$$\mathbb{P}\left(y_{n+n_{\mathrm{cal}}+1} \in C_\alpha(\boldsymbol{x}_{n+n_{\mathrm{cal}}+1})\right) \geq 1 - \alpha. \tag{2}$$

To obtain the guarantee equation (2), the only assumption required is that the calibration and test data form an exchangeable sequence (a condition weaker than, and therefore automatically satisfied by independence and identical distribution)Shafer & Vovk (2008) and that they are independent of the training data. It is essential to know that the guarantee equation (2) is marginal, i.e. holds in average over both the calibration dataset and the test sample choice. As we shall emphasize, there might be fluctuations due to the finite sample size of the calibration dataset.

## 3 RELATED WORKS

In this work, we study the potential of using Conformal Prediction as a statistical framework for interpreting OOD scores. This idea of casting OOD in a statistical framework has already been attempted in different settings.

**Selective Inference and Testing**  Selective Inference works on top of an ML predictor by using an additional decision function to decide for each example whether the original model's prediction should be considered. A score equivalent to an OOD score is used to define this decision function. Several approaches exist, for instance, through building a statistical test Haroush et al. (2022) or by training a neural network with an appropriate loss Geifman & El-Yaniv (2017; 2019). However, the framework of Conformal Prediction appears better suited to our goal since it applies to scores in a post-processing manner, does not require assumptions or modifications on the model, and benefits from dynamic development in the ML community.

**Conformal OOD and AD**  Conformal Prediction has been previously applied to Out-of-Distribution and Anomaly Detection. For instance, Liang et al. (2022) have proposed a method based on CP for OOD with labeled outliers, and Kaur et al. (2022) propose to use conformal p-values. CP is one of several frameworks that allow obtaining statistical guarantees for OOD detection. One of the first methods for Anomaly Detection was introduced by Vovk et al. (2003). Since then, several other methods have been proposed by Laxhammar & Falkman (2011); Laxhammar (2014);

Balasubramanian et al. (2014), as well as more recently Angelopoulos & Bates (2022); Guan & Tibshirani (2022), where the lengths of the prediction sets as OOD scores. These works all use the standard CP setting, in which basic marginal guarantees are obtained. We go further on this approach by using CP as a probabilistic tool to refine the interpretation and, hence, the usefulness of any OOD score.

**Finding Efficient Scores for Conformal Prediction**  We also investigate the benefits of using OOD scores as non-conformity scores in CP. Common ways to build prediction sets for classification, such as LAC Sadinle et al. (2019) or APS Romano et al. (2020) and RAPS Angelopoulos et al. (2020) are based on the softmax output of classifiers. However, non-conformity scores also exist for other predictors Vovk et al. (2005), for instance, based on nearest neighbor distance Shafer & Vovk (2008). In this work, we suggest interpreting any OOD score as a potential general replacement for scores in CP, opening a large avenue for CP score crafting. This idea could apply to any ML task, but we demonstrate that on a classification task, to be consistent with the standard OOD benchmark settings we follow in the present paper.

## 4   OOD Scores Through the Lens of CP

Let us begin by describing the typical benchmark setup for evaluating an OOD score. First, an OOD detector is fit on $\mathcal{D}_{id}^{train} = \{\boldsymbol{x}_1, \ldots, \boldsymbol{x}_n\}$. Then, the OOD score is evaluated on $\mathcal{D}_{id}^{val} = \{\boldsymbol{x}_{n+1}, \ldots, \boldsymbol{x}_{n+n_{\mathrm{val}}}\}$ and $\mathcal{D}_{ood} = \{\bar{\boldsymbol{x}}_1, \ldots, \bar{\boldsymbol{x}}_{n_{\mathrm{val}}}\}$, where $\mathcal{D}_{ood}$ is a dataset sampled from a different distribution $\mathcal{P}_{ood} \neq \mathcal{P}_{id}$ (typically, another dataset). We apply $s$ to obtain $\{s(\bar{\boldsymbol{x}}_1), \ldots, s(\bar{\boldsymbol{x}}_{n_{\mathrm{val}}}), s(\boldsymbol{x}_{n+1}), \ldots, s(\boldsymbol{x}_{n+n_{\mathrm{val}}})\}$. Then, we assess the discriminative power of $s$ by evaluating metrics depending on a threshold $\tau$. By considering ID samples as negative and OOD as positive, we can compute:

- The Area Under the Receiver Operating Characteristic (AUROC): we compute the False Positive Rate (FPR) and the True Positive Rate (TPR) for $\tau_i = s(\boldsymbol{x}_{n+i})$, $i \in \{1, \ldots, n_{\mathrm{val}}\}$, and compute the area under the curve with FPR as x-axis and TPR as y-axis.
- FPR@TPR95: The value of the False Positive Rate (FPR) when $\tau$ is selected among $\tau_1, \ldots, \tau_{n_{\mathrm{val}}}$ so that the True Positive Rate (TPR) is 0.95. It can be generalized to FPR@TPR$\beta$, for any $\beta \in (0, 1)$.

A crucial step in any of these metrics is to compute the FPR. The FPR and its empirical estimation $\widehat{\mathrm{FPR}}(\tau)$ are defined as follows:

$$\mathrm{FPR}(\tau) = \mathbb{P}_{\boldsymbol{x} \sim \mathcal{P}_{id}}(s(\boldsymbol{x}) \geq \tau), \qquad \widehat{\mathrm{FPR}}(\tau) = \frac{1}{n_{\mathrm{val}}} \sum_{i=1,\ldots,n_{\mathrm{val}}} \mathbf{1}_{s(\boldsymbol{x}_i) \geq \tau}. \tag{3}$$

### 4.1   OOD detection and p-values

Let us now rewrite the problem of OOD detection using the framework of statistical hypothesis testing. This framework allows us to reason in terms of p-values, which have multiple benefits: they have a rigorous mathematical definition and probabilistic interpretation, they can be interpreted equivalently for any score, and used for comparison of different scores. Given a test example $\boldsymbol{x}_{\mathrm{test}}$, we wish to test for $\boldsymbol{x}_{\mathrm{test}} \sim \mathcal{P}_{id}$, i.e. we wish to test the null hypothesis $\mathcal{H}_0 : \boldsymbol{x}_{\mathrm{test}} \sim \mathcal{P}_{id}$ against the alternate hypothesis $\mathcal{H}_1 : \boldsymbol{x}_{\mathrm{test}} \nsim \mathcal{P}_{id}$. The value $P_{\boldsymbol{x} \sim \mathcal{P}_{id}}(s(\boldsymbol{x}) \geq s(\boldsymbol{x}_{\mathrm{test}}))$ is an exact p-value for the null hypothesis $\mathcal{H}_0$. Note that this p-value corresponds to $\mathrm{FPR}(s(\boldsymbol{x}_{\mathrm{test}}))$ as defined in equation (3). Hence, the values $\widehat{\mathrm{FPR}}(\tau_1) = \widehat{\mathrm{FPR}}(s(\boldsymbol{x}_{n+1})), \ldots, \widehat{\mathrm{FPR}}(\tau_p) = \widehat{\mathrm{FPR}}(s(\boldsymbol{x}_{n+n_{\mathrm{val}}}))$ used in every OOD detection benchmark to compute the AUROC and FPR@TPR$\beta$ can be considered as approximate p-values. The relationship between the FPR and the p-values emphasizes the link between OOD detection evaluation and hypothesis testing.

### 4.2   Fluctuations of the p-value

This is where the framework of Conformal Prediction comes into play. Since we do not have access to the distribution $\mathcal{P}_{id}$, we approximate the FPRs (so the p-values) by using the validation dataset

$\mathcal{D}_{id}^{val}$, which allows using two results from CP to improve the evaluation of the FPR. Note that $\mathcal{D}_{id}^{val}$ can be related to the calibration dataset used in CP.

### 4.2.1 MARGINAL VALIDITY OF THE FPR

The first point that CP teaches us is that fluctuations in the scores of the validation dataset can lead to over-confident estimations of the p-value. In order to avoid that, we have to use the correction proposed by Bates et al. (2022) (which can be originally traced to Papadopoulos et al. (2002)):

$$\widehat{u}^{\mathrm{marg}}(\boldsymbol{x}) = \frac{1}{1 + n_{\mathrm{val}}} \left( 1 + \sum_{i=1\dots n_{\mathrm{val}}} \mathbf{1}_{s(\boldsymbol{x}_i) \geq s(\boldsymbol{x})} \right). \tag{4}$$

With this correction, if the $\boldsymbol{x}_i$ are i.i.d and the distribution of $s(\boldsymbol{x})$ under the ID law is continuous, we obtain *marginally valid* p-values, that is, p-values that satisfy

$$\mathbb{P}_{\boldsymbol{x} \sim \mathcal{P}_{id}}\big(\widehat{u}^{\mathrm{marg}}(\boldsymbol{x}) \leq t\big) \leq t, \quad \text{for all} \quad 0 \leq t \leq 1. \tag{5}$$

By *marginally*, we are pointing out that the probability in the above formula integrates over both the validation set $\mathcal{D}_{id}^{val}$ and the test point $\boldsymbol{x}$. This correction directly translates in terms of FPR. We can correct equation (3) to obtain a new estimation that enjoys this property:

$$\widehat{\mathrm{FPR}}(\tau) = \frac{1}{1 + n_{\mathrm{val}}} \left( 1 + \sum_{i=1\dots n_{\mathrm{val}}} \mathbf{1}_{s(\boldsymbol{x}_i) \geq \tau} \right). \tag{6}$$

However, the work of Bates et al. (2022) tells us that the FPR may still be overly confident. We discuss this point in the next section.

## 4.3 FLUCTUATIONS OF THE FPR

In this part, we mainly explain the work of Bates et al. (2022) that emphasizes that the FPR fluctuates depending on $\mathcal{D}_{id}^{val}$. We illustrate this phenomenon in the context of OOD detection and adapt the corrections proposed in Bates et al. (2022) to this field by defining new conformal AUROC and conformal FPR@TPR$\beta$.

Note first that the FPR can also be defined using a threshold $t$ applied to the p-values as:

$$\mathrm{FPR}(t, \mathcal{D}_{id}^{val}) = \mathbb{P}_{\boldsymbol{x} \sim \mathcal{P}_{id}}\big(\widehat{u}^{\mathrm{marg}}(\boldsymbol{x}) \leq t \,|\, \mathcal{D}_{id}^{val}\big), \tag{7}$$

where $t \in [0, 1]$. The authors point out that due to the empirical estimation of $\widehat{u}^{\mathrm{marg}}(\boldsymbol{x})$, the quantity $P_{\boldsymbol{x} \sim \mathcal{P}_{id}}\big(\widehat{u}^{\mathrm{marg}}(\boldsymbol{x}) \leq t \,|\, \mathcal{D}_{id}^{val}\big)$ is a random variable that depends on $\mathcal{D}_{id}^{val}$.

As a practical consequence, the FPR will fluctuate depending on which dataset $\mathcal{D}_{id}^{val}$ it is evaluated. The random variable $\mathrm{FPR}(t, \mathcal{D}_{id}^{val})$ follows a distribution that is known: it is a Beta distribution that depends on the parameters $n_{\mathrm{val}}$ and $t$:

$$\mathrm{FPR}(t, \mathcal{D}_{id}^{val}) \sim \mathrm{Beta}(\ell, n_{\mathrm{val}} + 1 - \ell), \tag{8}$$

where $\ell = \lfloor (n_{\mathrm{val}} + 1)t \rfloor$ (cf. Bates et al. (2022) or Vovk (2012) for a proof of the result).

### 4.3.1 ILLUSTRATION ON SVHN

To illustrate why this phenomenon matters in OOD detection, we leverage the fact that SVHN dataset provides an additional set of 530000 *extra* test images. It allows the simulation of 53 draws of the random variable $F(t; \mathcal{D}_{id}^{val})$, by splitting the over 530000 examples in the *svhn_extra* dataset into 53 different folds of 10000 examples each. For each fold, the 10000 examples are used to constitute the calibration dataset $\mathcal{D}_{id}^{val}$, whereas the remaining over 520000 examples are used to approximate the computation of $F$, i.e., given a calibration dataset $\mathcal{D}_{id}^{val}$,

$$F(t; \mathcal{D}_{id}^{val}) \approx \hat{F}(t; \mathcal{D}_{id}^{val}) = \frac{1}{520000} \sum_{i=1\dots520000} \mathbf{1}_{\widehat{u}^{\mathrm{marg}}(\boldsymbol{x}_i) \leq t}. \tag{9}$$

Due to the large number of points used in the approximating sum, the 53 values obtained are faithful approximations of the random variables $F(t; \mathcal{D}_{id}^{val})$.

We perform this simulation with $t = 0.1$ and plot the 53 values into a histogram. Additionally, we fit a Beta distribution to the histogram using the *scikit-learn* library. These plots are found in figure 1. As we can see, the estimated parameters of the fitted beta distribution are very close to those predicted by the theoretical result of equation (8). If the value $\widehat{u}^{\mathrm{marg}}(\boldsymbol{x})$ were a true p-value, the value of $F(t; \mathcal{D}_{id}^{val})$ would be equal to $\tau$, but as we can see from the theoretical result and the experiment above, $F(t; \mathcal{D}_{id}^{val})$ is a random variable that fluctuates around its mean value $\tau$. This phenomenon can be detrimental to safety-critical applications, which are the applications of choice for OOD detection. Indeed, it may result in underestimating the FPR, whereas we would like the FPR to be conservative.

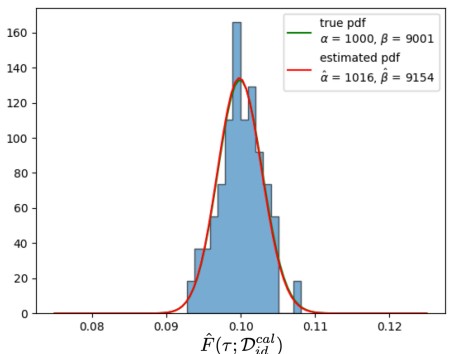

Figure 1: Histogram of $F(0.1; \mathcal{D}_{id}^{cal})$ for different calibration sets. The histogram is obtained by splitting the dataset *svhn_extra* into disjoint calibration sets of 10000 points each, and approximating the value of $F$ for each calibration set by integrating over the remaining 521131 examples.

### 4.3.2 PROBABILISTIC GUARANTEES FOR P-VALUES AND THE FPR

To solve this problem, Bates et al. (2022) further corrects the marginal p-values, thus obtaining *calibration-conditional* p-values. Given a user-predefined risk level $\delta$, the calibration-conditional p-values $\widehat{u}^{\mathrm{cc}}$ will satisfy

$$\mathbb{P}\Big(\mathbb{P}\big(\widehat{u}^{\mathrm{cc}}(\boldsymbol{x}) \leq t \,|\, \mathcal{D}_{id}^{val}\big) \leq t, \; \forall\, t \in (0,1)\Big) \geq 1 - \delta, \tag{10}$$

where the probability inside is taken over $\boldsymbol{x} \sim \mathcal{P}_{id}$, and the probability outside over the choice of $\mathcal{D}_{id}^{val}$. Thus, with a probability of at least $1 - \delta$, we can be confident that we have a *good* calibration set, meaning that our p-values will be conservative.

Likewise, we can correct the FPR directly. Bates et al. (2022) propose a correction of the empirical FPR that satisfies the following:

$$\mathbb{P}\left[\mathrm{FPR}(\tau) \leq \widehat{\mathrm{FPR}}^{+}(\tau), \forall \tau \in \mathbb{R}\right] \geq 1 - \delta, \tag{11}$$

where $\widehat{\mathrm{FPR}}^{+}(\tau)$ is a correction version of the empirical $\widehat{\mathrm{FPR}}(\tau)$. The corrected FPR is obtained by applying a correction function $h$ to the empirical FPR, i.e. $\widehat{\mathrm{FPR}}^{+}(\tau) = h \circ \widehat{\mathrm{FPR}}(\tau)$. In the following, we refer to the quantity $\widehat{\mathrm{FPR}}^{+}(\tau) = h \circ \widehat{\mathrm{FPR}}(\tau)$ as *conformal* FPR.

Four different correction functions $h$ are proposed by Bates et al. (2022), the Simes, DKWM, Asymptotic and Monte Carlo corrections. The Simes, DKWM and Monte Carlo corrections all provide the finite sample guarantees of equation (10) and equation (11), while the Asymptotic correction provides only an asymptotic guarantee, that is, when the number of calibration points goes to infinity. Between the three corrections providing the finite sample guarantee, we find the Monte Carlo one to give tighter bounds (please see Appendix A for more details on how the Simes and Monte Carlo corrections are defined).

### 4.4 CONFORMAL METRICS FOR OOD

Based on the previously defined conformal FPR (already defined in Bates et al. (2022), we define *conformal* AUROC and *conformal* FPR@TPR95. These two quantities are obtained similarly as their classical versions, but using the conformal FPR:

- Conformal AUROC: we compute the *conformal* FPR for $\tau_i = s(\boldsymbol{x}_{n+i})$, $i \in \{1, \ldots, n_{\mathrm{val}}\}$ and the True Positive Rate (TPR) for each of these values. We then compute the area under the curve with *conformal* FPR as x-axis and TPR as y-axis.

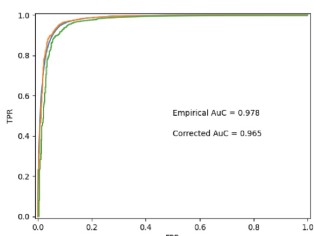 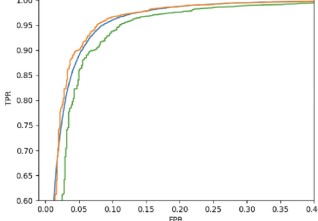 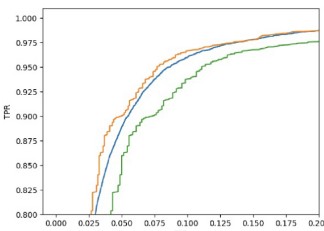

Figure 2: Different zoom levels of the ROC curves. The TPR is calculated by using all the points in the "Cifar10" dataset for the three curves. As for the TPR, the blue curve is obtained by using all data points in the "svhn_extra" dataset, the orange curve is an approximation of the blue curve using 1000 calibration points, whereas the green curve is obtained by correcting the FPR via the conformal AUROC method.

- Conformal FPR@TPR95: We select $\tau$ among $\tau_1, \ldots, \tau_{n_{\mathrm{val}}}$ so that the True Positive Rate (TPR) is $0.95$. We then compute the corresponding *conformal* FPR. It can be generalized to FPR@TPR$\beta$, for any $\beta \in (0, 1)$.

The computations are performed by considering the ID validation dataset as the calibration dataset. We would like to insist on the fact that Conformal FPR, AUROC, and FPR@TPR95 are not necessarily better approximations of the real FPR, AUROC and FPR@TPR95 values. Nonetheless, they are guaranteed to use conservative estimates of the FPR with a user-defined miscalibration tolerance $\delta$, which is an essential property in many safety-critical applications or certification processes Sellke et al. (2001). The effect of the correction on the ROC curve is illustrated in Figure 2 using the SVHN dataset as ID and Cifar-10 as OOD.

*Remark* 4.1 (Conformal metrics do not require extra validation data). Computing the conformal FPR only requires a correction to the estimated FPR. It does not require extra validation data. This is not like in CP, where we need a calibration dataset to find a threshold based on nonconformity scores obtained on calibration data, which is subsequently used to provide CP confidence intervals. Here, there are no confidence prediction intervals; we only use CP theory to obtain probabilistic guarantees of the FPR.

### 4.5 SAFER BENCHMARKS FOR OOD

AUROC and (to a lesser extent) FPR@TPR95 are two metrics that OOD and AD practitioners intensively use to benchmark and evaluate the performances of different OOD detection algorithms. However, as we saw in the previous sections, the evaluation can be overly optimistic, which can be detrimental to algorithms designed for safety-critical applications. In this section, we reevaluate various OOD baselines included in the very furnished OpenOOD Yang et al. (2022), and ADBench Han et al. (2022) benchmarks and illustrate the trade-off between performances and probabilistic guarantees. All our experiments can be easily carried out on a standard laptop CPU.

#### 4.5.1 OPENOOD

OpenOOD Yang et al. (2022) is an extensive benchmark for task-based OOD, i.e. for OOD methods that assess if some test data resembles some trained backbone's training data. Usually, backbones trained on CIFAR-10, CIFAR-100, Imagenet200, and Imagenet are considered. In our case, we consider a ResNet18 trained on the first three datasets only since we are not evaluating a new baseline but only investigating a new metric for the benchmark. We evaluate the AUROC of several baselines with various OOD datasets gathered into two groups, Near OOD and Far OOD, following OpenOOD's guidelines. We then compute the correction for the AUROC, with $\delta = 0.01$. The results are displayed in Table 1. We also run the benchmark for $\delta = 0.05$ and FPR-95, which we defer to Appendix C.

Table 1 shows that after the correction, the conformal AUROC is lower than the classical AUROC, by often more than 1 percent. On the one hand, this is significant, especially for such benchmarks where the State-of-the-art often holds by a fraction of a percentage. On the other hand, the correction is not *so* severe, and the best baselines still get very good AUROC despite the correction. In other

| | CIFAR-10 | | | | CIFAR-100 | | | | ImageNet-200 | | | |
|---|---|---|---|---|---|---|---|---|---|---|---|---|
| | Near OOD | | Far OOD | | Near OOD | | Far OOD | | Near OOD | | Far OOD | |
| OOD type | class. | conf. | class. | conf. | class. | conf. | class. | conf. | class. | conf. | class. | conf. |
| OpenMax Bendale & Boult (2015) | 87.2 | **85.95** | 89.53 | **88.3** | 76.66 | **74.95** | 79.12 | **77.52** | 80.4 | **78.82** | 90.41 | **88.77** |
| MSP Hendrycks & Gimpel (2018) | 87.68 | **86.56** | 91.0 | **89.98** | 80.42 | **78.93** | 77.58 | **76.0** | 83.3 | **81.85** | 90.2 | **88.83** |
| TempScale Guo et al. (2017) | 87.65 | **86.55** | 91.27 | **90.3** | 80.98 | **79.51** | 78.51 | **76.95** | 83.66 | **82.21** | 90.91 | **89.53** |
| ODIN Liang et al. (2018) | 80.25 | **79.04** | 87.21 | **86.26** | 79.8 | **78.3** | 79.44 | **77.92** | 80.32 | **78.85** | 91.89 | **90.59** |
| MDS Lee et al. (2018) | 86.72 | **85.49** | 90.2 | **89.09** | 58.79 | **56.85** | 70.06 | **68.31** | 62.51 | **60.68** | 74.94 | **73.09** |
| MDSEns Lee et al. (2018) | 60.46 | **58.69** | 74.07 | **72.72** | 45.98 | **43.97** | 66.03 | **64.43** | 54.58 | **52.76** | 70.08 | **68.35** |
| Gram Sastry & Oore (2020) | 52.63 | **50.69** | 69.74 | **68.11** | 50.69 | **48.69** | 73.97 | **72.63** | 68.36 | **66.74** | 70.94 | **69.3** |
| EBO Liu et al. (2020) | 86.93 | **85.9** | 91.74 | **90.9** | 80.84 | **79.36** | 79.71 | **78.19** | 82.57 | **81.1** | 91.12 | **89.71** |
| GradNorm Huang et al. (2021) | 53.77 | **51.92** | 58.55 | **56.76** | 69.73 | **68.11** | 68.82 | **67.19** | 73.33 | **71.85** | 85.29 | **83.99** |
| ReAct Sun et al. (2021) | 86.47 | **85.41** | 91.02 | **90.12** | 80.7 | **79.23** | 79.84 | **78.32** | 80.48 | **79.0** | 93.1 | **91.79** |
| MLS Hendrycks et al. (2022) | 86.86 | **85.81** | 91.61 | **90.74** | 81.04 | **79.58** | 79.6 | **78.07** | 82.96 | **81.5** | 91.34 | **89.94** |
| KLM Hendrycks et al. (2022) | 78.8 | **77.58** | 82.76 | **81.63** | 76.9 | **75.38** | 76.03 | **74.52** | 80.69 | **79.14** | 88.41 | **86.74** |
| VIM Wang et al. (2022) | 88.51 | **87.42** | 93.14 | **92.25** | 74.83 | **73.17** | 82.11 | **80.69** | 78.81 | **77.2** | 91.52 | **90.05** |
| KNN Sun et al. (2022) | 90.7 | **89.69** | 93.1 | **92.19** | 80.25 | **78.79** | 82.32 | **80.93** | 81.75 | **80.27** | 93.47 | **92.25** |
| DICE Sun & Li (2022) | 77.79 | **76.44** | 85.41 | **84.37** | 79.15 | **77.61** | 79.84 | **78.33** | 81.97 | **80.5** | 91.19 | **89.84** |
| RankFeat Song et al. (2022) | 76.33 | **74.76** | 70.15 | **68.39** | 62.22 | **60.33** | 67.74 | **65.9** | 58.57 | **57.0** | 38.97 | **37.09** |
| ASH Djurisic et al. (2022) | 74.11 | **72.71** | 78.36 | **77.02** | 78.39 | **76.89** | 79.7 | **78.23** | 82.12 | **80.72** | 94.23 | **93.11** |
| SHE Zhang et al. (2023) | 80.84 | **79.64** | 86.55 | **85.55** | 78.72 | **77.18** | 77.35 | **75.8** | 80.46 | **79.0** | 90.48 | **89.17** |

Table 1: Classical AUROC (class.) vs Conformal AUROC (conf.) obtained with the Monte Carlo method and $\delta = 0.01$ for several baselines from OpenOOD benchmark.

words, the correction is large enough to manifest its importance but low enough to still be useable in practice: **it costs only roughly** 1 **or** 2 **percent in AUROC to be** 99% **sure that the FPR involved in the AUROC calculation is not overestimated.**

### 4.5.2 ADBENCH

We perform the same procedure as OpenOOD with ADBench Han et al. (2022), which gathers many task-agnostic OOD baselines – considered Anomaly Detection (AD), hence the benchmark's name. We conduct the experiments with "unsupervised AD" baselines, i.e. baselines that do not leverage labeled anomalies. We apply the correction with $\delta = 0.05$ and summarize the results in Figure 3. The complete results are deferred to Appendix D.

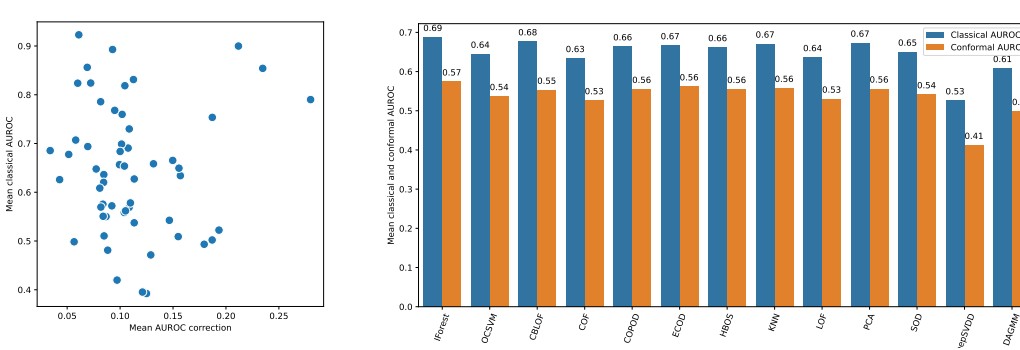

Figure 3: Results for ADBench benchmark. **(left)** Scatter plot with mean classical AUROC and mean AUROC correction over different methods for each dataset as y-axis and x-axis, respectively. **(right)** Mean AUROC and AUROC correction over different datasets for each AD method.

Figure 3 (left) shows a scatter plot with mean classical AUROC and mean AUROC correction over different methods for each dataset as y-axis and x-axis, respectively. The variability and magnitude of the correction are higher than for OpenOOD since the number of points in the test set changes depending on the dataset and is generally way lower. This observation is important because **it illustrates the brittleness of the conclusions that can be drawn from AD benchmarks and supports the increasingly commonly accepted fact that no method is provably better than others in AD** – one of the key conclusions of ADBench's paper itself Han et al. (2022). Figure 3 (right) shows the mean classical and conformal AUROC for each baseline over the datasets. The correction is more stable, demonstrating that the correction affects all baselines similarly.

## 5  OOD SCORES AS NONCONFORMITY SCORES FOR CP

In the previous sections, we have mostly emphasized that practitioners of OOD detection should look at CP as an additional building block for correctly interpreting the scores that all the OOD methods rely on. In this section, we advocate that the link between OOD and CP goes even deeper and that both fields could benefit from each other.

| | LAC | | | APS | | | RAPS | | |
|---|---|---|---|---|---|---|---|---|---|
| $\alpha$ | 0.005 | 0.01 | 0.05 | 0.005 | 0.01 | 0.05 | 0.005 | 0.01 | 0.05 |
| **Cifar10** | | | | | | | | | |
| Gram | 9.57 | 8.34 | 1.89 | $9.60 \pm 0.10$ | $1.93 \pm 0.06$ | $8.66 \pm 0.13$ | $9.56 \pm 0.13$ | $8.7 \pm 0.16$ | $1.89 \pm 0.03$ |
| ReAct | 3.75 | 1.98 | 1.03 | $4.47 \pm 0.16$ | $1.97 \pm 0.09$ | $3.62 \pm 0.17$ | $4.46 \pm 0.15$ | $3.67 \pm 0.19$ | $2.02 \pm 0.09$ |
| ODIN | 7.15 | 5.82 | 1.14 | $7.42 \pm 0.17$ | $1.53 \pm 0.06$ | $5.14 \pm 0.08$ | $7.45 \pm 0.16$ | $5.1 \pm 0.10$ | $1.57 \pm 0.08$ |
| KNN | 2.57 | **1.48** | **1.01** | $\underline{3.62} \pm 0.15$ | $\underline{1.09} \pm 0.03$ | $2.71 \pm 0.11$ | $\underline{3.69} \pm 0.11$ | $2.77 \pm 0.09$ | $\underline{1.08} \pm 0.02$ |
| Mahalanobis | **1.85** | $\underline{1.47}$ | 1.04 | $\mathbf{1.89} \pm 0.07$ | $\mathbf{1.04} \pm 0.01$ | $\mathbf{1.49} \pm 0.04$ | $\mathbf{1.92} \pm 0.05$ | $\mathbf{1.49} \pm 0.05$ | $\mathbf{1.04} \pm 0.01$ |
| CP (Softmax) | $\underline{2.44}$ | 1.73 | $\underline{1.03}$ | $3.92 \pm 0.26$ | $1.1 \pm 0.01$ | $\underline{2.16} \pm 0.13$ | $3.81 \pm 0.24$ | $\underline{2.17} \pm 0.11$ | $1.09 \pm 0.01$ |
| **Cifar100** | | | | | | | | | |
| ReAct | 52.41 | 29.77 | 10.06 | $53.02 \pm 0.27$ | $32.02 \pm 0.11$ | $10.45 \pm 0.15$ | $53.12 \pm 0.26$ | $32.12 \pm 0.13$ | $10.43 \pm 0.15$ |
| ODIN | 66.54 | 45.25 | 16.25 | $65.46 \pm 0.3$ | $45.56 \pm 0.14$ | $17.49 \pm 0.13$ | $65.61 \pm 0.25$ | $45.51 \pm 0.27$ | $17.6 \pm 0.14$ |
| KNN | 41.64 | 27.74 | 8.62 | $\underline{39.45} \pm 0.35$ | $29.81 \pm 0.24$ | $9.80 \pm 0.11$ | $\underline{39.63} \pm 0.21$ | $\underline{29.74} \pm 0.29$ | $\underline{9.81} \pm 0.10$ |
| Mahalanobis | **31.29** | **24.76** | $\underline{7.57}$ | $\mathbf{31.07} \pm 0.07$ | $\mathbf{24.77} \pm 0.20$ | $\mathbf{8.47} \pm 0.29$ | $\mathbf{31.15} \pm 0.07$ | $\mathbf{24.82} \pm 0.19$ | $\mathbf{8.48} \pm 0.21$ |
| CP (Softmax) | $\underline{31.96}$ | $\underline{27.21}$ | **5.73** | $46.55 \pm 1.47$ | $36.82 \pm 0.39$ | $17.59 \pm 0.41$ | $45.64 \pm 1.19$ | $36.83 \pm 0.79$ | $17.12 \pm 0.74$ |

Table 2: Efficiency (mean $\pm$ std. dev. for APS and RAPS) of the prediction sets for different scores for CP classification on CIFAR-10 and CIFAR-100. The best is bolded, the second is underlined.

So far, we have shown how OOD can use CP, but we argue that CP could also use OOD. Indeed, CP is about interpreting scores to provide probabilistic results. But CP works regardless of the given score. Indeed, all scores will have the same guarantee, but better scores will give tighter prediction sets, and worse scores will give very large and uninformative prediction sets. For CP to provide powerful probabilistic guarantees, the scores have to be informative, hence the common practice of relying on scores derived from the softmax values of a neural network Sadinle et al. (2019). It turns out that the maximum softmax is also a score used in OOD detection Hendrycks & Gimpel (2018), which suggests that OOD scores and CP scores might be related in some way. In this section, we explore using different OOD scores to perform CP. We consider two ResNet18 trained on CIFAR-10 and CIFAR-100 and build conformal prediction sets following the procedure described in section 2. To build these prediction sets, we use scores based on ReAct Sun et al. (2021), Gram Sastry & Oore (2020), KNN Sun et al. (2022), Mahalanobis Lee et al. (2018), and ODIN Liang et al. (2018). Note that we had to adapt those scores to make them class-dependent since the score used in CP is defined as $s_{cp}(\boldsymbol{x}, y)$. We did so following a procedure that we describe in detail in Appendix B. Then, given the OOD score $s(\boldsymbol{x}, y_i)$, we construct softmax-like scores $\hat{s}(\boldsymbol{x}, y_i) = \exp s(\boldsymbol{x}, y_i) / \sum_j \exp s(\boldsymbol{x}, y_j)$, and use it for CP.

For each defined score, we perform the calibration step on $n_{cal} = 2000$ points following the classical Least-Ambiguous set classifiers (LAC) procedure Sadinle et al. (2019), and the more recent Adaptive Prediction Set (APS) Romano et al. (2020) and Regularized Adaptive Prediction Set (RAPS) Angelopoulos et al. (2020) methods. For all methods, we construct the prediction sets for each of the remaining $n_{val} - n_{cal} = 8000$ points, and for coverages $1 - \alpha \in \{0.005, 0.01, 0.05\}$. We assess the mean efficiency of the prediction sets for each score, including LAC, APS, and RAPS based on softmax, as classically done in CP in Table 5). Since APS and RAPS involve sampling a uniform random variable, we report the mean and the standard deviation of the mean efficiency for 10 evaluations.

Table 5 shows that all OOD scores are inefficient for CP. For example, Gram performs very poorly (hence, we only run it on CIFAR-10). However, in some instances, some scores, like KNN or Mahalanobis, perform better than classical CP scores. This suggests that OOD scores may be good candidates as nonconformity scores.

## 6  LIMITATIONS

While we believe that OOD detection and CP have much to gain from each other, we acknowledge that our paper has limitations: *Data availability.* Computing conformal AUROC and conformal

FPR requires an extra calibration dataset, which might be a drawback in applications with low data availability. *Extra compute resources.* The extra calibration step requires additional calibration resources. However, these resources are negligible compared to those needed for training and fine-tuning a neural network.

# 7 CONCLUSION & DISCUSSION

In conclusion, our work highlights the inherent randomness of OOD metrics and demonstrates how Conformal Prediction (CP) can effectively correct these metrics. We have also shown that recent advancements in CP allow for uniform conservativeness guarantees on OOD metrics, providing more reliable evaluations. Furthermore, our analysis reveals that the correction introduced by CP does not significantly impact the performance of the best OOD baselines. On the other hand, we also showed that we could use OOD to improve existing CP techniques by using OOD scores as nonconformity scores. We found that some of them, especially Mahalanobis and KNN, are good candidates for nonconformity scores, unlocking a whole avenue for crafting CP nonconformity scores based on the plethora of existing post-hoc OOD scores.

By integrating CP with OOD, we have demonstrated the fruitful synergy between the two fields. OOD detection focuses on developing scores that accurately discriminate between OOD and ID, while CP specializes in interpreting scores to provide probabilistic guarantees. This interplay between OOD and CP presents opportunities for mutual advancement: advancements in CP research can enhance OOD by offering more refined probabilistic interpretations of OOD scores, which is particularly crucial in safety-critical applications. Conversely, progress in OOD research can benefit CP by providing scores that improve the efficiency of prediction sets. This suggests that further exploration and collaboration between the two fields hold great potential.

In summary, our findings underscore the intertwined nature of OOD and CP, emphasizing the need for continued investigation and cross-fertilization to advance both disciplines.

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

## A  APPENDIX: SIMES AND MONTE CARLO CORRECTIONS

In our work we use two of the corrections proposed by Bates et al. (2022), Simes and Monte Carlo Correction. In this section, we introduce these corrections for the sake of completeness, as well as two other corrections that we do not use for reasons to be detailed.

**Simes Correction**   Generally, we are interested in small p-values and the Simes correction focuses on those, that is by adding a smaller correction to the smaller p-values than the larger ones.

$$b_{n+1-i}^{\mathrm{s}} = 1 - \delta^{2/n} \left( \frac{i \cdots (i - n/2 + 1)}{n \cdots (n - n/2 + 1)} \right)^{2/n}, \quad i = 1, \ldots, n \tag{12}$$

**DKWM**   The former approach may be compared to the classical uniform concentration DKWM result, where the $b$ are defined as

$$b_i^{\mathrm{d}} = \min\{(i/n) + \sqrt{\log(2/\delta)/2n}, 1\}; \tag{13}$$

However, DKWM tends to provide much larger bounds than Simes.

**Asymptotic Correction**   The previous approach brought finite sample guarantees but at the cost of a large correction. In order to produce a tighter bound, for a more powerful test, we look into a correction that is correct asymptotically.

$$\begin{aligned} c_n(\delta) := \left( \sqrt{2 \log \log n} \right)^{-1} & \left( -\log[-\log(1 - \delta)] \right. \\ & \left. + 2 \log \log n + (1/2) \log \log \log n - (1/2) \log \pi \right). \end{aligned} \tag{14}$$

$$b_i^{\mathrm{a}} = \min \left\{ \frac{i}{n} + c_n(\delta) \frac{\sqrt{i(n-i)}}{n\sqrt{n}}, 1 \right\}, \quad i = 1, \ldots, n \tag{15}$$

This bound is quite similar to Simes for small values, but quite tighter for the remaining ones.

**Monte Carlo Correction**   The Monter Carlo Correction offers advantages of both the Simes and Asymptotic methods. It provides a finite-sample guarantee, mimics Simes for small p-values and remains closer to the asymptotic correction for larger ones.

$$h^{\mathrm{m}, \hat{\delta}}(t) = \min \left\{ h^{\mathrm{s}}(t), h^{\mathrm{a}, \hat{\delta}}(t) \right\}, \quad t \in [0, 1]. \tag{16}$$

# B  APPENDIX: DESIGNING CLASS-DEPENDENT OOD SCORES FOR CP

Let's consider a classification task with a classifier $f$ trained to fit a dataset $\{(\boldsymbol{x}_1, y_1), ..., (\boldsymbol{x}_n, y_n)\}$, where $\boldsymbol{x}_i \in \mathcal{X}$ and $y_i \in \{1, ...C\}$ for all $i \in \{1, ..., n\}$. In OOD, the score function $s : \mathcal{X} \to \mathbb{R}$, whereas in CP, the non-conformity score $s_{\text{cp}} : \mathcal{X} \times \mathbb{R} \to \mathbb{R}$. Hence, in order to construct a non-conformity score out of $s$, we have to make it class-dependent. In this section, we describe how to construct class-dependent OOD scores out of classical OOD scores for appropriate usage in CP.

## B.1  REACT

ReAct method Sun et al. (2021) gets the quantiles of $f$'s penultimate layer's activation values and then clips the activation values for a new input data point. The output softmax are then used for OOD scoring. Therefore, making the score class-dependent is straightforward: one only has to get the class softmax.

## B.2  ODIN

The idea of ODIN Liang et al. (2018) is also to tweak the network so that the softmax becomes more informative for OOD detection. Similarly to ReAct, one only has to get each class's softmax to make the score class-dependent.

## B.3  KNN

For each $\{\boldsymbol{x}_1, ..., \boldsymbol{x}_n\}$ from the training set, consider $\boldsymbol{H} = \{h(\boldsymbol{x}_1), ..., h(\boldsymbol{x}_n)\}$ where $h : \mathcal{X} \to \mathbb{R}^p$ is defined such that $h(\boldsymbol{x}_i)$ is the activation vector of $\boldsymbol{x}_i$ of $f$'s penultimate layer. Let $N_{\boldsymbol{H}} : \mathbb{R}^p \to \mathbb{R}^p$ be the nearest neighbor map such that $N(\boldsymbol{h})$ is the nearest neighbor of $\boldsymbol{h}$ among $\boldsymbol{H}$. KNN Sun et al. (2022) builds the score $s$ as

$$s(\boldsymbol{x}_{n+1}) = \|h(\boldsymbol{x}_{n+1}) - N_{\boldsymbol{H}}(h(\boldsymbol{x}_{n+1})\|.$$

To make this score class-dependent, one can build $C$ maps $\{N_{\boldsymbol{H}_1}, ..., N_{\boldsymbol{H}_C}\}$ where $\boldsymbol{H}_k = \{h(\boldsymbol{x}_i) | f(\boldsymbol{x}_i) = k\}$ and then define a new score

$$s(\boldsymbol{x}_{n+1}, y) = \|h(\boldsymbol{x}_{n+1}) - N_{\boldsymbol{H}_y}(h(\boldsymbol{x}_{n+1})\|$$

## B.4  MAHALANOBIS

Let consider the map $h$ as in KNN. For each $k \in \{1, ..., C\}$, Mahalanobis distance method Lee et al. (2018) computes $\Sigma_k$ and $\mu_k$, which are the empirical covariance matrix and mean vectors of each set of points $\{h(\boldsymbol{x}_i)\}_{i|f(\boldsymbol{x}_i)=k}$. Then, the score $s$ is computed as:

$$s(\boldsymbol{x}_{n+1}) = \sqrt{(\boldsymbol{x}_{n+1} - \mu_{f(\boldsymbol{x}_{n+1})})^T \Sigma^{-1} (\boldsymbol{x}_{n+1} - \mu_{f(\boldsymbol{x}_{n+1})})},$$

where $\Sigma = \frac{1}{C} \sum_{k \in \{1,...,C\}} \Sigma_k$. To make the score class-dependent, one simply has to define

$$s(\boldsymbol{x}_{n+1}, y) = \sqrt{(\boldsymbol{x}_{n+1} - \mu_y)^T \Sigma_y^{-1} (\boldsymbol{x}_{n+1} - \mu_y)}.$$

## B.5  GRAM

Let $f$ be a classifier of depth $L$. Gram method Sastry & Oore (2020) builds a statistic $\delta : \mathcal{X} \to \mathbb{R}^L$ that outputs the channel-wise correlation of the activation maps for each layer. First, $\{\delta(\boldsymbol{x}_1), ..., \delta(\boldsymbol{x}_n)\}$ are computed. Then, a multi-dimensional statistic $\{d_{l,k}\}_{l \in \{1,...,L\}, k \in \{1,...,C\}}$ is computed for each layer after a class-wise aggregation.

For a new test point $\boldsymbol{x}_{n+1}$, $\delta(\boldsymbol{x}_{n+1})$ is computed, along with $f(\boldsymbol{x}_{n+1})$. The score is built out of a weighted mean of the layer-wise deviation:

$$s(\boldsymbol{x}_{n+1}) = \sum_{l \in \{1,...,L\}} w_l |\delta(\boldsymbol{x}_{n+1})_l - d_{l,f(\boldsymbol{x}_{n+1})}|,$$

where $\{w_l\}_{l\in\{1,\dots,L\}}$ are some normalization weights computed with the training data. It is quite straightforward to make this OOD score class-dependent by defining

$$s(\boldsymbol{x}_{n+1}, y) = \sum_{l\in\{1,\dots,L\}} w_l |\delta(\boldsymbol{x}_{n+1})_l - d_{l,y}|.$$

## C  APPENDIX: COMPLEMENTARY RESULTS ON OPENOOD BENCHMARK

In this section, we present the full results of benchmarks on OpenOOD. The results displayed are AUROC with $\delta = 0.05$ in Table 3, FPR@TPR95 with $\delta = 0.05$ in Table 5 and FPR@TPR95 with $\delta = 0.01$ in Table 4.

| | CIFAR-10 | | | | CIFAR-100 | | | | ImageNet-200 | | | |
|---|---|---|---|---|---|---|---|---|---|---|---|---|
| | Near | | Far | | Near | | Far | | Near | | Far | |
| OOD type | marg. | conf. | marg. | conf. | marg. | conf. | marg. | conf. | marg. | conf. | marg. | conf. |
| OpenMax Bendale & Boult (2015) | 87.2 | **86.18** | 89.53 | **88.52** | 76.66 | **75.26** | 79.12 | **77.81** | 80.4 | **79.2** | 90.41 | **89.49** |
| MSP Hendrycks & Gimpel (2016) | 87.68 | **86.76** | 91.0 | **90.17** | 80.42 | **79.2** | 77.58 | **76.29** | 83.3 | **82.2** | 90.2 | **89.36** |
| TempScale Guo et al. (2017) | 87.65 | **86.75** | 91.27 | **90.48** | 80.98 | **79.78** | 78.51 | **77.24** | 83.66 | **82.58** | 90.91 | **90.11** |
| ODIN Liang et al. (2018) | 80.25 | **79.26** | 87.21 | **86.43** | 79.8 | **78.57** | 79.44 | **78.2** | 80.32 | **79.19** | 91.89 | **91.17** |
| MDS Lee et al. (2018) | 86.72 | **85.71** | 90.2 | **89.29** | 58.79 | **57.2** | 70.06 | **68.63** | 62.51 | **60.96** | 74.94 | **73.6** |
| MDSEns Lee et al. (2018) | 60.46 | **59.01** | 74.07 | **72.96** | 45.98 | **44.34** | 66.03 | **64.72** | 54.58 | **52.99** | 70.08 | **68.76** |
| Gram Sastry & Oore (2020) | 52.63 | **51.04** | 69.74 | **68.41** | 50.69 | **49.06** | 73.97 | **72.87** | 68.36 | **67.0** | 70.94 | **69.69** |
| EBO Liu et al. (2020) | 86.93 | **86.08** | 91.74 | **91.05** | 80.84 | **79.63** | 79.71 | **78.47** | 82.57 | **81.47** | 91.12 | **90.33** |
| GradNorm Huang et al. (2021) | 53.77 | **52.26** | 58.55 | **57.09** | 69.73 | **68.41** | 68.82 | **67.48** | 73.33 | **72.12** | 85.29 | **84.45** |
| ReAct Sun et al. (2021) | 86.47 | **85.6** | 91.02 | **90.28** | 80.7 | **79.5** | 79.84 | **78.6** | 80.48 | **79.35** | 93.1 | **92.4** |
| MLS Hendrycks et al. (2022) | 86.86 | **86.0** | 91.61 | **90.9** | 81.04 | **79.84** | 79.6 | **78.35** | 82.96 | **81.88** | 91.34 | **90.56** |
| KLM Hendrycks et al. (2022) | 78.8 | **77.8** | 82.76 | **81.83** | 76.9 | **75.65** | 76.03 | **74.8** | 80.69 | **79.54** | 88.41 | **87.44** |
| VIM Wang et al. (2022) | 88.51 | **87.62** | 93.14 | **92.41** | 74.83 | **73.47** | 82.11 | **80.95** | 78.81 | **77.57** | 91.52 | **90.7** |
| KNN Sun et al. (2022) | 90.7 | **89.87** | 93.1 | **92.35** | 80.25 | **79.05** | 82.32 | **81.19** | 81.75 | **80.63** | 93.47 | **92.83** |
| DICE Sun & Li (2022) | 77.79 | **76.68** | 85.41 | **84.56** | 79.15 | **77.89** | 79.84 | **78.64** | 81.97 | **80.86** | 91.19 | **90.43** |
| RankFeat Song et al. (2022) | 76.33 | **75.05** | 70.15 | **68.71** | 62.22 | **60.67** | 67.74 | **66.24** | 58.57 | **57.06** | 38.97 | **37.43** |
| ASH Djurisic et al. (2022) | 74.11 | **72.96** | 78.36 | **77.27** | 78.39 | **77.16** | 79.7 | **78.5** | 82.12 | **81.07** | 94.23 | **93.66** |
| SHE Zhang et al. (2023) | 80.84 | **79.86** | 86.55 | **85.73** | 78.72 | **77.46** | 77.35 | **76.08** | 80.46 | **79.34** | 90.48 | **89.72** |

Table 3: Classical AUROC (marg.) vs Conformal AUROC (conf.) obtained with the Monte Carlo method and $\delta = 0.05$ for several baselines from OpenOOD benchmark.

| | CIFAR-10 | | | | CIFAR-100 | | | | ImageNet-200 | | | |
|---|---|---|---|---|---|---|---|---|---|---|---|---|
| | Near | | Far | | Near | | Far | | Near | | Far | |
| OOD type | marg. | conf. | marg. | conf. | marg. | conf. | marg. | conf. | marg. | conf. | marg. | conf. |
| OpenMax Bendale & Boult (2015) | 46.77 | **48.98** | 29.48 | **31.48** | 55.57 | **57.8** | 54.77 | **57.0** | 63.32 | **65.75** | 32.29 | **35.35** |
| MSP Hendrycks & Gimpel (2016) | 53.57 | **55.8** | 31.44 | **33.45** | 54.73 | **56.96** | 59.08 | **61.31** | 55.25 | **57.69** | 35.44 | **38.29** |
| TempScale Guo et al. (2017) | 56.85 | **59.08** | 33.36 | **35.38** | 54.77 | **56.99** | 58.24 | **60.47** | 55.03 | **57.5** | 34.11 | **37.06** |
| ODIN Liang et al. (2018) | 84.55 | **86.78** | 60.9 | **62.97** | 58.44 | **60.67** | 57.75 | **59.98** | 66.38 | **68.8** | 33.66 | **36.75** |
| MDS Lee et al. (2018) | 46.22 | **48.44** | 30.3 | **32.3** | 82.75 | **84.98** | 70.46 | **72.68** | 79.34 | **81.52** | 61.26 | **63.81** |
| MDSEns Lee et al. (2018) | 92.06 | **94.29** | 61.09 | **62.87** | 95.84 | **98.07** | 66.97 | **68.85** | 91.69 | **93.8** | 80.43 | **82.89** |
| Gram Sastry & Oore (2020) | 93.52 | **95.75** | 69.29 | **71.48** | 92.48 | **94.71** | 63.1 | **65.2** | 85.43 | **87.63** | 84.95 | **87.44** |
| EBO Liu et al. (2020) | 67.54 | **69.77** | 40.55 | **42.58** | 55.49 | **57.72** | 56.41 | **58.64** | 59.46 | **61.93** | 34.0 | **37.07** |
| GradNorm Huang et al. (2021) | 95.37 | **97.6** | 89.34 | **91.52** | 86.13 | **88.36** | 82.79 | **85.02** | 83.07 | **85.33** | 66.78 | **69.67** |
| ReAct Sun et al. (2021) | 71.56 | **73.78** | 42.43 | **44.52** | 56.74 | **58.97** | 56.32 | **58.55** | 65.37 | **67.8** | 27.21 | **30.28** |
| MLS Hendrycks et al. (2022) | 67.54 | **69.77** | 40.53 | **42.56** | 55.48 | **57.71** | 56.53 | **58.76** | 58.94 | **61.44** | 33.59 | **36.68** |
| KLM Hendrycks et al. (2022) | 86.41 | **88.63** | 76.42 | **78.65** | 79.52 | **81.75** | 70.16 | **72.39** | 69.42 | **71.91** | 39.57 | **42.56** |
| VIM Wang et al. (2022) | 48.07 | **50.29** | 25.77 | **27.65** | 62.96 | **65.19** | 49.72 | **51.95** | 59.91 | **62.32** | 26.86 | **29.81** |
| KNN Sun et al. (2022) | 34.54 | **36.65** | 23.88 | **25.77** | 61.32 | **63.54** | 54.04 | **56.27** | 60.42 | **62.9** | 26.49 | **29.66** |
| DICE Sun & Li (2022) | 80.15 | **82.38** | 53.93 | **56.06** | 58.1 | **60.33** | 55.95 | **58.17** | 60.98 | **63.46** | 35.93 | **39.04** |
| RankFeat Song et al. (2022) | 67.38 | **69.61** | 68.24 | **70.47** | 79.94 | **82.17** | 68.89 | **71.11** | 92.02 | **93.91** | 98.48 | **99.58** |
| ASH Djurisic et al. (2022) | 89.03 | **91.26** | 76.66 | **78.89** | 66.14 | **68.37** | 62.67 | **64.89** | 65.95 | **68.44** | 26.26 | **29.46** |
| SHE Zhang et al. (2023) | 84.49 | **86.72** | 63.26 | **65.41** | 59.32 | **61.54** | 62.74 | **64.97** | 65.92 | **68.31** | 41.5 | **44.62** |

Table 4: Classical FPR@TPR95 (marg.) vs Conformal FPR@TPR95 (conf.) obtained with the Monte Carlo method and $\delta = 0.01$ for several baselines from OpenOOD benchmark.

| OOD type | CIFAR-10 | | | | CIFAR-100 | | | | ImageNet-200 | | | |
|---|---|---|---|---|---|---|---|---|---|---|---|---|
| | Near | | Far | | Near | | Far | | Near | | Far | |
| | marg. | conf. | marg. | conf. | marg. | conf. | marg. | conf. | marg. | conf. | marg. | conf. |
| OpenMax Bendale & Boult (2015) | 46.77 | **48.58** | 29.48 | **31.11** | 55.57 | **57.39** | 54.77 | **56.59** | 63.32 | **65.14** | 32.29 | **33.98** |
| MSP Hendrycks & Gimpel (2016) | 53.57 | **55.39** | 31.44 | **33.08** | 54.73 | **56.55** | 59.08 | **60.9** | 55.25 | **57.06** | 35.44 | **37.16** |
| TempScale Guo et al. (2017) | 56.85 | **58.67** | 33.36 | **35.01** | 54.77 | **56.59** | 58.24 | **60.06** | 55.03 | **56.85** | 34.11 | **35.8** |
| ODIN Liang et al. (2018) | 84.55 | **86.37** | 60.9 | **62.59** | 58.44 | **60.26** | 57.75 | **59.57** | 66.38 | **68.2** | 33.66 | **35.34** |
| MDS Lee et al. (2018) | 46.22 | **48.03** | 30.3 | **31.94** | 82.75 | **84.57** | 70.46 | **72.28** | 79.34 | **81.16** | 61.26 | **63.08** |
| MDSEns Lee et al. (2018) | 92.06 | **93.88** | 61.09 | **62.54** | 95.84 | **97.66** | 66.97 | **68.5** | 91.69 | **93.51** | 80.43 | **82.25** |
| Gram Sastry & Oore (2020) | 93.52 | **95.34** | 69.29 | **71.08** | 92.48 | **94.3** | 63.1 | **64.81** | 85.43 | **87.25** | 84.95 | **86.77** |
| EBO Liu et al. (2020) | 67.54 | **69.36** | 40.55 | **42.21** | 55.49 | **57.31** | 56.41 | **58.23** | 59.46 | **61.28** | 34.0 | **35.7** |
| GradNorm Huang et al. (2021) | 95.37 | **97.19** | 89.34 | **91.16** | 86.13 | **87.95** | 82.79 | **84.61** | 83.07 | **84.89** | 66.78 | **68.6** |
| ReAct Sun et al. (2021) | 71.56 | **73.38** | 42.43 | **44.14** | 56.74 | **58.56** | 56.32 | **58.14** | 65.37 | **67.19** | 27.21 | **28.81** |
| MLS Hendrycks et al. (2022) | 67.54 | **69.36** | 40.53 | **42.19** | 55.48 | **57.3** | 56.53 | **58.35** | 58.94 | **60.76** | 33.59 | **35.28** |
| KLM Hendrycks et al. (2022) | 86.41 | **88.23** | 76.42 | **78.24** | 79.52 | **81.34** | 70.16 | **71.98** | 69.42 | **71.24** | 39.57 | **41.3** |
| VIM Wang et al. (2022) | 48.07 | **49.88** | 25.77 | **27.3** | 62.96 | **64.78** | 49.72 | **51.54** | 59.91 | **61.72** | 26.86 | **28.46** |
| KNN Sun et al. (2022) | 34.54 | **36.27** | 23.88 | **25.42** | 61.32 | **63.14** | 54.04 | **55.86** | 60.42 | **62.23** | 26.49 | **28.09** |
| DICE Sun & Li (2022) | 80.15 | **81.97** | 53.93 | **55.67** | 58.1 | **59.92** | 55.95 | **57.77** | 60.98 | **62.8** | 35.93 | **37.66** |
| RankFeat Song et al. (2022) | 67.38 | **69.2** | 68.24 | **70.06** | 79.94 | **81.76** | 68.89 | **70.71** | 92.02 | **93.84** | 98.48 | **99.55** |
| ASH Djurisic et al. (2022) | 89.03 | **90.85** | 76.66 | **78.48** | 66.14 | **67.96** | 62.67 | **64.49** | 65.95 | **67.77** | 26.26 | **27.85** |
| SHE Zhang et al. (2023) | 84.49 | **86.31** | 63.26 | **65.02** | 59.32 | **61.14** | 62.74 | **64.56** | 65.92 | **67.74** | 41.5 | **43.27** |

Table 5: Classical FPR@TPR95 (marg.) vs Conformal FPR@TPR95 (conf.) obtained with the Monte Carlo method and $\delta = 0.05$ for several baselines from OpenOOD benchmark.

## D  APPENDIX: FULL RESULTS FOR ADBENCH

In this section, we present the full results of the ADBench benchmark. Table 6 displays classical AUROC, Table 7 displays conformal AUROC, and Table 8 displays the difference between the two (AUROC correction), all with $\delta = 0.05$.

| | IForest | OCSVM | CBLOF | COF | COPOD | ECOD | HBOS | KNN | LOF | PCA | SOD | DeepSVDD | DAGMM |
|---|---|---|---|---|---|---|---|---|---|---|---|---|---|
| cover | 0.87 | 0.93 | 0.89 | 0.77 | 0.89 | 0.92 | 0.80 | 0.86 | 0.85 | 0.94 | 0.74 | 0.46 | 0.90 |
| donors | 0.78 | 0.72 | 0.62 | 0.71 | 0.82 | 0.89 | 0.78 | 0.82 | 0.59 | 0.83 | 0.56 | 0.36 | 0.71 |
| fault | 0.57 | 0.48 | 0.64 | 0.62 | 0.44 | 0.45 | 0.51 | 0.73 | 0.59 | 0.46 | 0.68 | 0.52 | 0.46 |
| fraud | 0.90 | 0.91 | 0.88 | 0.96 | 0.88 | 0.89 | 0.90 | 0.93 | 0.96 | 0.90 | 0.95 | 0.73 | 0.90 |
| glass | 0.77 | 0.35 | 0.83 | 0.72 | 0.72 | 0.66 | 0.77 | 0.82 | 0.69 | 0.66 | 0.73 | 0.47 | 0.76 |
| Hepatitis | 0.70 | 0.68 | 0.66 | 0.41 | 0.82 | 0.75 | 0.80 | 0.53 | 0.38 | 0.76 | 0.68 | 0.52 | 0.55 |
| Ionosphere | 0.84 | 0.76 | 0.91 | 0.87 | 0.79 | 0.73 | 0.62 | 0.88 | 0.91 | 0.79 | 0.86 | 0.51 | 0.73 |
| landsat | 0.48 | 0.36 | 0.64 | 0.53 | 0.42 | 0.36 | 0.55 | 0.58 | 0.54 | 0.36 | 0.60 | 0.63 | 0.44 |
| ALOI | 0.57 | 0.56 | 0.55 | 0.65 | 0.54 | 0.56 | 0.53 | 0.61 | 0.67 | 0.57 | 0.61 | 0.51 | 0.52 |
| letter | 0.61 | 0.46 | 0.76 | 0.80 | 0.54 | 0.56 | 0.60 | 0.86 | 0.84 | 0.50 | 0.84 | 0.56 | 0.50 |
| 20news 0 | 0.64 | 0.63 | 0.71 | 0.71 | 0.61 | 0.61 | 0.62 | 0.73 | 0.80 | 0.64 | 0.73 | 0.50 | 0.63 |
| 20news 1 | 0.51 | 0.53 | 0.52 | 0.58 | 0.52 | 0.54 | 0.53 | 0.57 | 0.61 | 0.54 | 0.58 | 0.48 | 0.54 |
| 20news 2 | 0.50 | 0.51 | 0.47 | 0.53 | 0.50 | 0.52 | 0.51 | 0.51 | 0.54 | 0.51 | 0.50 | 0.49 | 0.53 |
| 20news 3 | 0.75 | 0.72 | 0.83 | 0.81 | 0.75 | 0.75 | 0.74 | 0.79 | 0.71 | 0.73 | 0.70 | 0.67 | 0.54 |
| 20news 4 | 0.48 | 0.51 | 0.45 | 0.57 | 0.48 | 0.51 | 0.50 | 0.48 | 0.51 | 0.51 | 0.53 | 0.53 | 0.48 |
| 20news 5 | 0.52 | 0.49 | 0.47 | 0.50 | 0.48 | 0.46 | 0.49 | 0.48 | 0.55 | 0.48 | 0.48 | 0.49 | 0.54 |
| Lymphography | 1.00 | 1.00 | 1.00 | 0.91 | 0.99 | 1.00 | 0.99 | 0.56 | 0.90 | 1.00 | 0.73 | 0.34 | 0.72 |
| magic.gamma | 0.73 | 0.61 | 0.75 | 0.67 | 0.68 | 0.64 | 0.71 | 0.82 | 0.69 | 0.67 | 0.75 | 0.60 | 0.59 |
| musk | 1.00 | 0.81 | 1.00 | 0.39 | 0.94 | 0.95 | 1.00 | 0.70 | 0.41 | 1.00 | 0.74 | 0.56 | 0.77 |
| PageBlocks | 0.90 | 0.89 | 0.85 | 0.73 | 0.88 | 0.92 | 0.81 | 0.82 | 0.76 | 0.91 | 0.78 | 0.59 | 0.90 |
| pendigits | 0.95 | 0.94 | 0.90 | 0.45 | 0.91 | 0.93 | 0.93 | 0.73 | 0.48 | 0.94 | 0.66 | 0.42 | 0.64 |
| Pima | 0.73 | 0.67 | 0.71 | 0.61 | 0.69 | 0.63 | 0.71 | 0.73 | 0.66 | 0.71 | 0.61 | 0.51 | 0.56 |
| annthyroid | 0.82 | 0.57 | 0.62 | 0.66 | 0.77 | 0.79 | 0.60 | 0.72 | 0.70 | 0.66 | 0.77 | 0.77 | 0.57 |
| satellite | 0.70 | 0.59 | 0.71 | 0.55 | 0.63 | 0.58 | 0.75 | 0.65 | 0.56 | 0.60 | 0.64 | 0.55 | 0.62 |
| satimage-2 | 0.99 | 0.97 | 1.00 | 0.57 | 0.97 | 0.96 | 0.98 | 0.93 | 0.47 | 0.98 | 0.83 | 0.49 | 0.96 |
| shuttle | 1.00 | 0.97 | 0.83 | 0.52 | 0.99 | 0.99 | 0.99 | 0.70 | 0.57 | 0.99 | 0.70 | 0.49 | 0.98 |
| smtp | 0.86 | 0.72 | 0.70 | 0.69 | 0.70 | 0.78 | 0.56 | 0.84 | 0.58 | 0.83 | 0.40 | 0.72 | 0.71 |
| speech | 0.51 | 0.50 | 0.51 | 0.56 | 0.53 | 0.51 | 0.51 | 0.51 | 0.52 | 0.51 | 0.56 | 0.54 | 0.53 |
| Stamps | 0.91 | 0.84 | 0.68 | 0.54 | 0.93 | 0.88 | 0.91 | 0.69 | 0.51 | 0.91 | 0.73 | 0.56 | 0.89 |
| thyroid | 0.98 | 0.88 | 0.95 | 0.91 | 0.94 | 0.98 | 0.96 | 0.96 | 0.87 | 0.96 | 0.93 | 0.49 | 0.80 |
| vertebral | 0.37 | 0.38 | 0.41 | 0.49 | 0.26 | 0.41 | 0.29 | 0.34 | 0.49 | 0.37 | 0.40 | 0.37 | 0.53 |
| vowels | 0.75 | 0.63 | 0.90 | 0.95 | 0.55 | 0.62 | 0.73 | 0.97 | 0.93 | 0.67 | 0.92 | 0.56 | 0.61 |
| Waveform | 0.71 | 0.56 | 0.72 | 0.73 | 0.75 | 0.62 | 0.69 | 0.74 | 0.73 | 0.65 | 0.69 | 0.56 | 0.49 |
| WDBC | 0.99 | 0.99 | 0.99 | 0.96 | 0.99 | 0.97 | 0.99 | 0.92 | 0.89 | 0.99 | 0.92 | 0.62 | 0.77 |
| Wilt | 0.42 | 0.31 | 0.33 | 0.50 | 0.33 | 0.36 | 0.32 | 0.48 | 0.51 | 0.20 | 0.53 | 0.46 | 0.37 |
| wine | 0.80 | 0.73 | 0.26 | 0.44 | 0.89 | 0.77 | 0.91 | 0.45 | 0.38 | 0.84 | 0.46 | 0.60 | 0.62 |
| WPBC | 0.47 | 0.45 | 0.45 | 0.46 | 0.49 | 0.47 | 0.51 | 0.47 | 0.41 | 0.46 | 0.51 | 0.50 | 0.48 |
| yeast | 0.38 | 0.41 | 0.45 | 0.44 | 0.37 | 0.44 | 0.40 | 0.39 | 0.45 | 0.41 | 0.42 | 0.48 | 0.41 |
| campaign | 0.73 | 0.67 | 0.64 | 0.58 | 0.78 | 0.77 | 0.79 | 0.73 | 0.59 | 0.73 | 0.69 | 0.53 | 0.58 |
| cardio | 0.93 | 0.94 | 0.90 | 0.71 | 0.92 | 0.94 | 0.85 | 0.77 | 0.66 | 0.96 | 0.73 | 0.58 | 0.75 |
| Cardiotocography | 0.68 | 0.78 | 0.65 | 0.54 | 0.67 | 0.78 | 0.61 | 0.56 | 0.60 | 0.75 | 0.52 | 0.53 | 0.62 |
| celeba | 0.70 | 0.71 | 0.74 | 0.39 | 0.76 | 0.76 | 0.76 | 0.60 | 0.39 | 0.79 | 0.48 | 0.54 | 0.45 |
| CIFAR10 0 | 0.73 | 0.68 | 0.70 | 0.70 | 0.69 | 0.70 | 0.70 | 0.74 | 0.74 | 0.70 | 0.71 | 0.56 | 0.53 |
| CIFAR10 1 | 0.55 | 0.59 | 0.61 | 0.63 | 0.46 | 0.51 | 0.44 | 0.60 | 0.72 | 0.60 | 0.62 | 0.50 | 0.58 |
| CIFAR10 2 | 0.56 | 0.58 | 0.58 | 0.61 | 0.56 | 0.57 | 0.54 | 0.60 | 0.65 | 0.58 | 0.59 | 0.58 | 0.51 |
| CIFAR10 3 | 0.55 | 0.58 | 0.59 | 0.56 | 0.51 | 0.53 | 0.50 | 0.56 | 0.60 | 0.56 | 0.56 | 0.60 | 0.56 |
| CIFAR10 5 | 0.50 | 0.58 | 0.58 | 0.57 | 0.47 | 0.52 | 0.47 | 0.54 | 0.60 | 0.57 | 0.54 | 0.46 | 0.59 |
| CIFAR10 6 | 0.64 | 0.65 | 0.68 | 0.69 | 0.65 | 0.66 | 0.65 | 0.72 | 0.72 | 0.68 | 0.69 | 0.57 | 0.50 |
| CIFAR10 7 | 0.54 | 0.59 | 0.56 | 0.57 | 0.52 | 0.55 | 0.50 | 0.54 | 0.60 | 0.57 | 0.56 | 0.62 | 0.61 |
| agnews 0 | 0.50 | 0.47 | 0.54 | 0.61 | 0.49 | 0.47 | 0.48 | 0.58 | 0.63 | 0.47 | 0.56 | 0.35 | 0.48 |
| agnews 1 | 0.58 | 0.54 | 0.58 | 0.71 | 0.51 | 0.54 | 0.55 | 0.62 | 0.74 | 0.55 | 0.61 | 0.37 | 0.56 |
| agnews 2 | 0.65 | 0.61 | 0.71 | 0.73 | 0.61 | 0.59 | 0.61 | 0.75 | 0.79 | 0.61 | 0.73 | 0.50 | 0.53 |
| agnews 3 | 0.54 | 0.55 | 0.57 | 0.70 | 0.51 | 0.53 | 0.51 | 0.62 | 0.70 | 0.55 | 0.61 | 0.50 | 0.51 |
| amazon | 0.56 | 0.54 | 0.58 | 0.58 | 0.57 | 0.54 | 0.56 | 0.59 | 0.56 | 0.54 | 0.58 | 0.45 | 0.51 |
| imdb | 0.50 | 0.45 | 0.50 | 0.49 | 0.50 | 0.45 | 0.48 | 0.48 | 0.49 | 0.46 | 0.50 | 0.52 | 0.42 |
| yelp | 0.61 | 0.59 | 0.64 | 0.68 | 0.60 | 0.57 | 0.59 | 0.68 | 0.66 | 0.59 | 0.66 | 0.50 | 0.55 |

Table 6: Full results for ADBench: classical AUROC.

| | IForest | OCSVM | CBLOF | COF | COPOD | ECOD | HBOS | KNN | LOF | PCA | SOD | DeepSVDD | DAGMM |
|---|---|---|---|---|---|---|---|---|---|---|---|---|---|
| cover | 0.75 | 0.83 | 0.79 | 0.64 | 0.78 | 0.82 | 0.74 | 0.74 | 0.73 | 0.84 | 0.60 | 0.30 | 0.79 |
| donors | 0.72 | 0.66 | 0.54 | 0.64 | 0.77 | 0.85 | 0.72 | 0.76 | 0.53 | 0.78 | 0.50 | 0.31 | 0.65 |
| fault | 0.48 | 0.40 | 0.55 | 0.53 | 0.35 | 0.37 | 0.43 | 0.65 | 0.50 | 0.37 | 0.59 | 0.43 | 0.37 |
| fraud | 0.77 | 0.63 | 0.63 | 0.82 | 0.62 | 0.64 | 0.86 | 0.68 | 0.79 | 0.66 | 0.69 | 0.51 | 0.64 |
| glass | 0.65 | 0.31 | 0.73 | 0.60 | 0.60 | 0.53 | 0.64 | 0.72 | 0.58 | 0.54 | 0.63 | 0.38 | 0.66 |
| Hepatitis | 0.54 | 0.52 | 0.52 | 0.23 | 0.70 | 0.61 | 0.66 | 0.26 | 0.22 | 0.62 | 0.54 | 0.38 | 0.40 |
| Ionosphere | 0.77 | 0.68 | 0.85 | 0.80 | 0.70 | 0.63 | 0.50 | 0.83 | 0.85 | 0.71 | 0.80 | 0.38 | 0.64 |
| landsat | 0.42 | 0.30 | 0.58 | 0.48 | 0.35 | 0.30 | 0.49 | 0.52 | 0.48 | 0.30 | 0.54 | 0.58 | 0.39 |
| ALOI | 0.49 | 0.46 | 0.45 | 0.55 | 0.43 | 0.46 | 0.46 | 0.52 | 0.58 | 0.47 | 0.52 | 0.41 | 0.42 |
| letter | 0.44 | 0.30 | 0.61 | 0.66 | 0.36 | 0.39 | 0.42 | 0.74 | 0.72 | 0.33 | 0.71 | 0.40 | 0.35 |
| 20news 0 | 0.51 | 0.49 | 0.57 | 0.60 | 0.48 | 0.46 | 0.47 | 0.62 | 0.69 | 0.50 | 0.61 | 0.36 | 0.49 |
| 20news 1 | 0.36 | 0.39 | 0.37 | 0.44 | 0.37 | 0.39 | 0.37 | 0.44 | 0.47 | 0.39 | 0.44 | 0.33 | 0.39 |
| 20news 2 | 0.34 | 0.36 | 0.34 | 0.39 | 0.34 | 0.36 | 0.35 | 0.36 | 0.38 | 0.36 | 0.34 | 0.32 | 0.38 |
| 20news 3 | 0.65 | 0.61 | 0.73 | 0.71 | 0.63 | 0.64 | 0.64 | 0.69 | 0.58 | 0.62 | 0.59 | 0.54 | 0.45 |
| 20news 4 | 0.28 | 0.31 | 0.27 | 0.39 | 0.27 | 0.32 | 0.30 | 0.30 | 0.34 | 0.31 | 0.35 | 0.35 | 0.30 |
| 20news 5 | 0.34 | 0.32 | 0.29 | 0.30 | 0.30 | 0.31 | 0.32 | 0.29 | 0.35 | 0.32 | 0.28 | 0.31 | 0.35 |
| Lymphography | 0.95 | 0.95 | 0.95 | 0.83 | 0.95 | 0.95 | 0.95 | 0.45 | 0.81 | 0.96 | 0.62 | 0.25 | 0.62 |
| magic.gamma | 0.70 | 0.57 | 0.72 | 0.63 | 0.65 | 0.60 | 0.68 | 0.79 | 0.65 | 0.64 | 0.72 | 0.56 | 0.55 |
| musk | 0.57 | 0.65 | 0.22 | 0.21 | 0.83 | 0.85 | 0.58 | 0.53 | 0.22 | 0.32 | 0.59 | 0.42 | 0.64 |
| PageBlocks | 0.84 | 0.83 | 0.79 | 0.67 | 0.82 | 0.86 | 0.74 | 0.76 | 0.70 | 0.85 | 0.71 | 0.52 | 0.84 |
| pendigits | 0.87 | 0.86 | 0.82 | 0.33 | 0.82 | 0.85 | 0.85 | 0.60 | 0.35 | 0.86 | 0.53 | 0.29 | 0.52 |
| Pima | 0.63 | 0.57 | 0.62 | 0.51 | 0.59 | 0.54 | 0.62 | 0.64 | 0.55 | 0.61 | 0.52 | 0.40 | 0.45 |
| annthyroid | 0.76 | 0.49 | 0.55 | 0.59 | 0.70 | 0.72 | 0.53 | 0.65 | 0.63 | 0.59 | 0.71 | 0.71 | 0.49 |
| satellite | 0.67 | 0.55 | 0.67 | 0.50 | 0.59 | 0.54 | 0.71 | 0.61 | 0.51 | 0.56 | 0.59 | 0.50 | 0.58 |
| satimage-2 | 0.51 | 0.71 | 0.45 | 0.41 | 0.74 | 0.80 | 0.77 | 0.79 | 0.34 | 0.70 | 0.68 | 0.31 | 0.84 |
| shuttle | 0.94 | 0.87 | 0.74 | 0.46 | 0.89 | 0.94 | 0.94 | 0.65 | 0.52 | 0.85 | 0.63 | 0.42 | 0.91 |
| smtp | 0.81 | 0.60 | 0.58 | 0.59 | 0.58 | 0.68 | 0.45 | 0.76 | 0.48 | 0.72 | 0.32 | 0.60 | 0.60 |
| speech | 0.31 | 0.31 | 0.31 | 0.37 | 0.34 | 0.32 | 0.31 | 0.31 | 0.33 | 0.32 | 0.35 | 0.34 | 0.34 |
| Stamps | 0.84 | 0.75 | 0.57 | 0.42 | 0.87 | 0.80 | 0.83 | 0.57 | 0.39 | 0.85 | 0.62 | 0.42 | 0.81 |
| thyroid | 0.90 | 0.77 | 0.86 | 0.81 | 0.86 | 0.90 | 0.92 | 0.87 | 0.77 | 0.88 | 0.84 | 0.33 | 0.69 |
| vertebral | 0.23 | 0.26 | 0.28 | 0.37 | 0.14 | 0.29 | 0.17 | 0.20 | 0.37 | 0.25 | 0.29 | 0.24 | 0.40 |
| vowels | 0.57 | 0.45 | 0.73 | 0.76 | 0.35 | 0.45 | 0.54 | 0.77 | 0.74 | 0.49 | 0.75 | 0.35 | 0.43 |
| Waveform | 0.56 | 0.42 | 0.59 | 0.58 | 0.60 | 0.47 | 0.53 | 0.59 | 0.50 | 0.54 | 0.39 | 0.34 |
| WDBC | 0.95 | 0.95 | 0.95 | 0.91 | 0.95 | 0.92 | 0.96 | 0.86 | 0.81 | 0.95 | 0.85 | 0.50 | 0.67 |
| Wilt | 0.29 | 0.20 | 0.21 | 0.37 | 0.20 | 0.24 | 0.19 | 0.35 | 0.38 | 0.12 | 0.42 | 0.34 | 0.26 |
| wine | 0.70 | 0.63 | 0.12 | 0.31 | 0.80 | 0.67 | 0.84 | 0.33 | 0.26 | 0.75 | 0.32 | 0.48 | 0.48 |
| WPBC | 0.33 | 0.32 | 0.32 | 0.33 | 0.36 | 0.33 | 0.38 | 0.32 | 0.29 | 0.33 | 0.39 | 0.38 | 0.35 |
| yeast | 0.29 | 0.31 | 0.35 | 0.35 | 0.28 | 0.34 | 0.31 | 0.30 | 0.36 | 0.31 | 0.32 | 0.38 | 0.31 |
| campaign | 0.68 | 0.62 | 0.59 | 0.52 | 0.74 | 0.72 | 0.75 | 0.68 | 0.53 | 0.68 | 0.64 | 0.48 | 0.52 |
| cardio | 0.84 | 0.85 | 0.80 | 0.60 | 0.83 | 0.84 | 0.75 | 0.67 | 0.53 | 0.86 | 0.62 | 0.47 | 0.64 |
| Cardiotocography | 0.59 | 0.70 | 0.57 | 0.45 | 0.58 | 0.70 | 0.52 | 0.48 | 0.51 | 0.66 | 0.43 | 0.45 | 0.53 |
| celeba | 0.64 | 0.65 | 0.69 | 0.30 | 0.70 | 0.71 | 0.71 | 0.28 | 0.30 | 0.74 | 0.38 | 0.48 | 0.37 |
| CIFAR10 0 | 0.63 | 0.59 | 0.60 | 0.60 | 0.59 | 0.60 | 0.60 | 0.65 | 0.64 | 0.61 | 0.61 | 0.46 | 0.42 |
| CIFAR10 1 | 0.43 | 0.48 | 0.50 | 0.52 | 0.35 | 0.39 | 0.33 | 0.49 | 0.62 | 0.49 | 0.51 | 0.39 | 0.48 |
| CIFAR10 2 | 0.45 | 0.48 | 0.47 | 0.50 | 0.44 | 0.45 | 0.43 | 0.49 | 0.55 | 0.47 | 0.48 | 0.48 | 0.40 |
| CIFAR10 3 | 0.44 | 0.48 | 0.49 | 0.46 | 0.40 | 0.42 | 0.39 | 0.47 | 0.50 | 0.46 | 0.46 | 0.49 | 0.45 |
| CIFAR10 5 | 0.38 | 0.48 | 0.47 | 0.46 | 0.35 | 0.40 | 0.34 | 0.42 | 0.49 | 0.46 | 0.42 | 0.34 | 0.49 |
| CIFAR10 6 | 0.54 | 0.55 | 0.58 | 0.59 | 0.54 | 0.55 | 0.54 | 0.61 | 0.62 | 0.58 | 0.59 | 0.47 | 0.39 |
| CIFAR10 7 | 0.43 | 0.48 | 0.45 | 0.46 | 0.41 | 0.44 | 0.39 | 0.44 | 0.50 | 0.46 | 0.46 | 0.51 | 0.50 |
| agnews 0 | 0.41 | 0.39 | 0.45 | 0.53 | 0.40 | 0.38 | 0.39 | 0.49 | 0.56 | 0.39 | 0.48 | 0.27 | 0.40 |
| agnews 1 | 0.50 | 0.46 | 0.50 | 0.64 | 0.42 | 0.45 | 0.46 | 0.54 | 0.67 | 0.47 | 0.53 | 0.28 | 0.48 |
| agnews 2 | 0.57 | 0.53 | 0.63 | 0.66 | 0.52 | 0.51 | 0.52 | 0.68 | 0.73 | 0.53 | 0.66 | 0.42 | 0.45 |
| agnews 3 | 0.45 | 0.46 | 0.49 | 0.63 | 0.43 | 0.44 | 0.43 | 0.54 | 0.64 | 0.46 | 0.53 | 0.42 | 0.42 |
| amazon | 0.48 | 0.45 | 0.50 | 0.49 | 0.48 | 0.46 | 0.47 | 0.50 | 0.48 | 0.46 | 0.50 | 0.37 | 0.43 |
| imdb | 0.41 | 0.36 | 0.41 | 0.40 | 0.42 | 0.36 | 0.40 | 0.39 | 0.40 | 0.37 | 0.41 | 0.44 | 0.34 |
| yelp | 0.52 | 0.50 | 0.55 | 0.60 | 0.52 | 0.49 | 0.51 | 0.60 | 0.59 | 0.51 | 0.58 | 0.42 | 0.47 |

Table 7: Full results for ADBench: conformal AUROC.

| | IForest | OCSVM | CBLOF | COF | COPOD | ECOD | HBOS | KNN | LOF | PCA | SOD | DeepSVDD | DAGMM |
|---|---|---|---|---|---|---|---|---|---|---|---|---|---|
| cover | 0.12 | 0.10 | 0.11 | 0.13 | 0.11 | 0.10 | 0.07 | 0.12 | 0.12 | 0.10 | 0.14 | 0.15 | 0.11 |
| donors | 0.06 | 0.07 | 0.08 | 0.07 | 0.05 | 0.04 | 0.06 | 0.06 | 0.06 | 0.05 | 0.06 | 0.05 | 0.06 |
| fault | 0.09 | 0.08 | 0.09 | 0.09 | 0.09 | 0.09 | 0.09 | 0.08 | 0.09 | 0.09 | 0.09 | 0.08 | 0.09 |
| fraud | 0.13 | 0.28 | 0.25 | 0.14 | 0.26 | 0.25 | 0.04 | 0.26 | 0.16 | 0.25 | 0.26 | 0.22 | 0.26 |
| glass | 0.12 | 0.05 | 0.10 | 0.12 | 0.12 | 0.13 | 0.13 | 0.10 | 0.11 | 0.13 | 0.10 | 0.09 | 0.10 |
| Hepatitis | 0.16 | 0.15 | 0.14 | 0.18 | 0.12 | 0.14 | 0.13 | 0.27 | 0.16 | 0.14 | 0.14 | 0.14 | 0.14 |
| Ionosphere | 0.08 | 0.08 | 0.06 | 0.07 | 0.09 | 0.10 | 0.12 | 0.05 | 0.05 | 0.08 | 0.06 | 0.13 | 0.09 |
| landsat | 0.06 | 0.06 | 0.05 | 0.05 | 0.06 | 0.06 | 0.06 | 0.06 | 0.05 | 0.06 | 0.05 | 0.05 | 0.05 |
| ALOI | 0.07 | 0.10 | 0.10 | 0.09 | 0.10 | 0.10 | 0.06 | 0.09 | 0.08 | 0.10 | 0.09 | 0.10 | 0.10 |
| letter | 0.17 | 0.16 | 0.15 | 0.14 | 0.19 | 0.17 | 0.18 | 0.13 | 0.13 | 0.17 | 0.13 | 0.15 | 0.15 |
| 20news 0 | 0.14 | 0.14 | 0.13 | 0.11 | 0.14 | 0.15 | 0.14 | 0.11 | 0.11 | 0.14 | 0.12 | 0.13 | 0.14 |
| 20news 1 | 0.15 | 0.15 | 0.15 | 0.14 | 0.15 | 0.15 | 0.16 | 0.13 | 0.14 | 0.15 | 0.14 | 0.14 | 0.15 |
| 20news 2 | 0.16 | 0.15 | 0.14 | 0.15 | 0.16 | 0.16 | 0.16 | 0.15 | 0.16 | 0.15 | 0.15 | 0.16 | 0.15 |
| 20news 3 | 0.10 | 0.11 | 0.10 | 0.10 | 0.11 | 0.11 | 0.10 | 0.10 | 0.13 | 0.11 | 0.12 | 0.13 | 0.09 |
| 20news 4 | 0.20 | 0.20 | 0.17 | 0.18 | 0.21 | 0.19 | 0.20 | 0.18 | 0.16 | 0.20 | 0.17 | 0.18 | 0.18 |
| 20news 5 | 0.17 | 0.17 | 0.18 | 0.20 | 0.18 | 0.15 | 0.17 | 0.19 | 0.20 | 0.16 | 0.20 | 0.18 | 0.19 |
| Lymphography | 0.05 | 0.05 | 0.05 | 0.08 | 0.04 | 0.05 | 0.05 | 0.11 | 0.09 | 0.04 | 0.11 | 0.09 | 0.11 |
| magic.gamma | 0.03 | 0.04 | 0.03 | 0.04 | 0.03 | 0.04 | 0.03 | 0.03 | 0.04 | 0.03 | 0.03 | 0.04 | 0.04 |
| musk | 0.43 | 0.15 | 0.78 | 0.18 | 0.11 | 0.11 | 0.42 | 0.17 | 0.19 | 0.68 | 0.16 | 0.14 | 0.13 |
| PageBlocks | 0.06 | 0.06 | 0.06 | 0.06 | 0.06 | 0.06 | 0.07 | 0.06 | 0.06 | 0.06 | 0.07 | 0.06 | 0.05 |
| pendigits | 0.08 | 0.08 | 0.09 | 0.12 | 0.08 | 0.08 | 0.08 | 0.13 | 0.13 | 0.08 | 0.13 | 0.13 | 0.12 |
| Pima | 0.10 | 0.10 | 0.10 | 0.10 | 0.10 | 0.09 | 0.09 | 0.10 | 0.10 | 0.09 | 0.10 | 0.11 | 0.11 |
| annthyroid | 0.06 | 0.08 | 0.07 | 0.07 | 0.07 | 0.06 | 0.07 | 0.07 | 0.07 | 0.07 | 0.06 | 0.06 | 0.08 |
| satellite | 0.04 | 0.04 | 0.04 | 0.05 | 0.04 | 0.04 | 0.04 | 0.05 | 0.05 | 0.04 | 0.05 | 0.05 | 0.04 |
| satimage-2 | 0.48 | 0.26 | 0.55 | 0.16 | 0.23 | 0.16 | 0.21 | 0.14 | 0.13 | 0.27 | 0.15 | 0.18 | 0.12 |
| shuttle | 0.06 | 0.10 | 0.09 | 0.06 | 0.10 | 0.05 | 0.05 | 0.04 | 0.05 | 0.13 | 0.06 | 0.07 | 0.07 |
| smtp | 0.05 | 0.12 | 0.12 | 0.10 | 0.12 | 0.11 | 0.10 | 0.09 | 0.09 | 0.11 | 0.08 | 0.12 | 0.11 |
| speech | 0.19 | 0.19 | 0.19 | 0.19 | 0.19 | 0.19 | 0.19 | 0.20 | 0.19 | 0.19 | 0.21 | 0.21 | 0.18 |
| Stamps | 0.07 | 0.09 | 0.11 | 0.11 | 0.06 | 0.08 | 0.07 | 0.12 | 0.12 | 0.07 | 0.11 | 0.14 | 0.07 |
| thyroid | 0.08 | 0.10 | 0.09 | 0.10 | 0.08 | 0.08 | 0.04 | 0.09 | 0.10 | 0.08 | 0.09 | 0.16 | 0.11 |
| vertebral | 0.14 | 0.12 | 0.13 | 0.12 | 0.12 | 0.12 | 0.12 | 0.14 | 0.13 | 0.12 | 0.11 | 0.12 | 0.13 |
| vowels | 0.19 | 0.18 | 0.17 | 0.20 | 0.20 | 0.17 | 0.20 | 0.20 | 0.19 | 0.18 | 0.17 | 0.21 | 0.19 |
| Waveform | 0.15 | 0.15 | 0.14 | 0.15 | 0.15 | 0.15 | 0.15 | 0.14 | 0.14 | 0.16 | 0.15 | 0.16 | 0.16 |
| WDBC | 0.04 | 0.04 | 0.04 | 0.06 | 0.04 | 0.05 | 0.04 | 0.06 | 0.08 | 0.04 | 0.07 | 0.12 | 0.10 |
| Wilt | 0.13 | 0.11 | 0.12 | 0.12 | 0.13 | 0.12 | 0.14 | 0.13 | 0.13 | 0.08 | 0.12 | 0.12 | 0.11 |
| wine | 0.10 | 0.10 | 0.14 | 0.14 | 0.09 | 0.11 | 0.08 | 0.12 | 0.11 | 0.10 | 0.15 | 0.11 | 0.13 |
| WPBC | 0.13 | 0.13 | 0.13 | 0.13 | 0.13 | 0.13 | 0.14 | 0.14 | 0.12 | 0.13 | 0.12 | 0.12 | 0.13 |
| yeast | 0.09 | 0.10 | 0.10 | 0.10 | 0.09 | 0.09 | 0.09 | 0.09 | 0.10 | 0.10 | 0.11 | 0.10 | 0.10 |
| campaign | 0.05 | 0.05 | 0.05 | 0.06 | 0.05 | 0.05 | 0.04 | 0.05 | 0.06 | 0.05 | 0.06 | 0.05 | 0.06 |
| cardio | 0.09 | 0.09 | 0.10 | 0.12 | 0.09 | 0.09 | 0.10 | 0.10 | 0.14 | 0.10 | 0.11 | 0.11 | 0.11 |
| Cardiotocography | 0.09 | 0.08 | 0.08 | 0.09 | 0.09 | 0.08 | 0.08 | 0.08 | 0.09 | 0.09 | 0.08 | 0.08 | 0.09 |
| celeba | 0.06 | 0.06 | 0.05 | 0.09 | 0.05 | 0.05 | 0.05 | 0.31 | 0.09 | 0.05 | 0.10 | 0.06 | 0.08 |
| CIFAR10 0 | 0.10 | 0.10 | 0.10 | 0.10 | 0.10 | 0.10 | 0.10 | 0.10 | 0.10 | 0.10 | 0.10 | 0.10 | 0.11 |
| CIFAR10 1 | 0.12 | 0.11 | 0.11 | 0.11 | 0.11 | 0.12 | 0.11 | 0.11 | 0.10 | 0.11 | 0.10 | 0.10 | 0.11 |
| CIFAR10 2 | 0.11 | 0.11 | 0.11 | 0.11 | 0.11 | 0.11 | 0.11 | 0.11 | 0.11 | 0.11 | 0.11 | 0.10 | 0.11 |
| CIFAR10 3 | 0.11 | 0.11 | 0.10 | 0.10 | 0.11 | 0.11 | 0.11 | 0.10 | 0.10 | 0.11 | 0.10 | 0.11 | 0.11 |
| CIFAR10 5 | 0.12 | 0.10 | 0.11 | 0.11 | 0.12 | 0.12 | 0.12 | 0.12 | 0.11 | 0.11 | 0.12 | 0.12 | 0.10 |
| CIFAR10 6 | 0.11 | 0.10 | 0.11 | 0.10 | 0.11 | 0.11 | 0.11 | 0.10 | 0.10 | 0.10 | 0.10 | 0.10 | 0.11 |
| CIFAR10 7 | 0.11 | 0.11 | 0.10 | 0.10 | 0.11 | 0.11 | 0.11 | 0.10 | 0.10 | 0.10 | 0.10 | 0.11 | 0.11 |
| agnews 0 | 0.09 | 0.09 | 0.08 | 0.08 | 0.09 | 0.09 | 0.09 | 0.08 | 0.08 | 0.09 | 0.08 | 0.09 | 0.09 |
| agnews 1 | 0.09 | 0.09 | 0.08 | 0.07 | 0.09 | 0.09 | 0.09 | 0.08 | 0.07 | 0.09 | 0.08 | 0.09 | 0.09 |
| agnews 2 | 0.08 | 0.08 | 0.08 | 0.07 | 0.08 | 0.09 | 0.08 | 0.07 | 0.06 | 0.08 | 0.07 | 0.08 | 0.08 |
| agnews 3 | 0.09 | 0.09 | 0.08 | 0.07 | 0.09 | 0.09 | 0.09 | 0.08 | 0.07 | 0.09 | 0.08 | 0.08 | 0.08 |
| amazon | 0.08 | 0.08 | 0.08 | 0.08 | 0.08 | 0.08 | 0.08 | 0.09 | 0.09 | 0.08 | 0.09 | 0.08 | 0.08 |
| imdb | 0.09 | 0.09 | 0.09 | 0.09 | 0.09 | 0.09 | 0.09 | 0.09 | 0.09 | 0.09 | 0.09 | 0.08 | 0.08 |
| yelp | 0.08 | 0.08 | 0.08 | 0.08 | 0.08 | 0.09 | 0.08 | 0.08 | 0.08 | 0.08 | 0.08 | 0.08 | 0.08 |

Table 8: Full results for ADBench: AUROC correction (difference between conformal and classical AUROC).

