# OpenReview forum: "Exploring the Link Between Out-of-Distribution Detection and Conformal Prediction with Illustrations of Its Benefits"
_ICLR.cc/2025/Conference — Submitted to ICLR 2025_

### Official Review · Reviewer_u4qk · 2024-10-16

**Soundness:** 3
**Presentation:** 3
**Contribution:** 3
**Rating:** 6
**Confidence:** 3

**Summary:**

The authors ask themselves, and answer positively, to the question whether CP can be beneficial to OOD, and whether the vice versa holds as well. They also validate empirically their findings.

**Strengths:**

The paper makes an extremely interesting parallel between OOD and CP. In addition, I found very interesting the marginal guarantees that the authors were able to find.

**Weaknesses:**

CP is a method for *uncertainty representation*, not uncertainty *quantification*. Indeed, CP *represents* uncertainty via the conformal prediction region. It does not quantify it: there is no real value attached to any kind of predictive uncertainty (e.g. aleatoric or epistemic, AU and EU, respectively). Some claim that the diameter of the conformal prediction region quantifies the uncertainty, but even in that case, it is unable to distinguish between AU and EU. Indeed, the diameter is a positive function of both: it increases as both increase, and hence it cannot be used to distinguish between the two [1]. Please add this clarification in the camera-ready version of the manuscript.

The simplest CP technique is (arguably) transductive CP, not split CP.

Shouldn't $n_\text{val}$ in (2) and in (10) be substituted by $P$? In particular, CP guarantees hold for all exchangeable countably additive probabilities $P$ on the space $\mathcal{Y}$ of outputs.

How does the proposed method relate to the subsequent work by Kaur [2]?

---

[1] https://arxiv.org/abs/2302.09656

[2] https://arxiv.org/abs/2302.11019

**Questions:**

See Weaknesses. In addition, Typo at line 69: eas (2023). Shouldn't it be Easa?

---

> ### Author Response · Authors · 2024-11-22
>
> We would like to thank the reviewer u4qk for recognizing the importance of drawing the parallel between OOD and CP. We address your remarks in the following.
>
> - **On uncertainty representation:** Thank you for this interesting remark, we include it in the revised version of the manuscript and add that CP is also a method for uncertainty representation, supported by the reference you provided (thank you!). Still, we also leave that it is an uncertainty quantification technique, because it is broadly recognized as such by the community.
> - We corrected the typo and corrected the statement about split CP being the simplest approach (now we say that it is one of the simplest approaches)
> - **About the work of Kaur [2]**: After careful reading, we do not see connections with Kaur [2] other than the fact that they propose a new OOD detection score and a redefinition of OOD - which is related to one of the research fields of our work (OOD) but, in our opinion, not to our contributions. But we are curious about what you have in mind and would be glad to know about the link that you see if you see any!

---

> > ### Comment · Reviewer_u4qk · 2024-11-26
> > **Thank you**
> >
> > I'm happy with the author response, and I'm happy to keep my score

---

> > > ### Author Response · Authors · 2024-11-28
> > >
> > > Thank you for taking the time to read our response; we are happy to have successfully comforted your initial judgment.

---

### Official Review · Reviewer_16UK · 2024-10-30

**Soundness:** 3
**Presentation:** 3
**Contribution:** 3
**Rating:** 6
**Confidence:** 3

**Summary:**

This paper explores the connection between Out-of-Distribution (OOD) detection scores and the non-conformity scores in Conformal Prediction (CP), showing the potential for cross-fertilization between these methods. Non-conformity scores offer probabilistic interpretation and correction for OOD scores, while OOD scores can enhance the efficiency of prediction sets obtained through CP.

**Strengths:**

•	OOD scores can be unreliable, and employing new statistical developments to improve their reliability is a promising direction. Using the uncertainty estimates from CP appears to be a robust approach to address OOD score unreliability.

•	The paper targets a practical problem, effectively demonstrating key concepts through comprehensive case studies.

•	By linking OOD detection with CP, this work facilitates cross-fertilization that benefits both machine learning and statistics. The authors provide a good introduction to using CP for interpreting and correcting OOD scores.

**Weaknesses:**

•	Non-conformity scores in CP require an evaluation dataset, and it reliability depends on the choice of this dataset. This introduces a practical challenge in implementing this statistically sound solution.

•	The paper presents limited innovation, as much of the work is based on existing research.

**Questions:**

I think that there are two types of uncertainties for a model predicting a new test instance:

1.	Prediction Confidence: Assuming the instance is within the In-distribution, this type of uncertainty relates to the strength of evidence supporting the prediction.

2.	In-Distribution Uncertainty: As discussed in this paper, this refers to uncertainty regarding whether the instance truly belongs in the In-distribution.

I would appreciate further discussion from the authors on the distinctions and connections between these two types of uncertainties. Can they be quantified into a single metric?

---

> ### Author Response · Authors · 2024-11-22
>
> We would like to thank reviewer 16UK for acknowledging the importance of providing probabilistic guarantees to evaluation metrics, especially for safety-critical applications. We appreciate that you grasped the importance of linking CP and OOD. We first address your main concern and then answer the question section.
>
> ---
> # Main concerns
>
> ## 1. The contributions are not limited to the corrected metrics
>
> This criticism is supported by the observation that the technical foundations of our corrected metrics rely on Bates et al. (2022). This observation is true, and we acknowledge it several times in the manuscript.  However, deriving corrected metrics is far from being the only contribution of this work. In addition to this contribution:
>
> - We point out that the common practice of OOD and Anomaly detection enables the use of this theoretical work. Drawing this link is a contribution.
> - We also use OOD scores to build new prediction intervals, which is **not related to the work of Bates et al. (2022)**. It opens a whole avenue for nonconformity score crafting, which is central to CP performances.
> - To us, perhaps the most important contribution is the **message** carried by the paper, supported by our several contributions : OOD and CP are closely related and should be used jointly. We believe that this message could have a significant impact on both fields, which are very dynamic and active research fields in AI.
>
> We agree that each contribution taken independently would not be a sufficient contribution for a whole paper. But they form a coherent whole that advocates for further exploration of the link between CP and OOD, which is, in our opinion, a very important insight and could have a **broad impact on future research in both fields**.
>
> ## 2. We do not need extra validation (or calibration) data
>
> We should have insisted more on that point because it is another advantage of our method: **it does not require extra validation data**.
>
> Computing the conformal FPR only requires a correction on the estimated FPR that does not rely on a third calibration dataset. It is not like in CP where we need a calibration dataset to find a threshold based on nonconformity scores obtained on calibration data, that is subsequently used to provide CP confidence intervals. Here, there are no confidence prediction intervals; we only use CP theory to obtain probabilistic guarantees on the FPR. The steps are as follows:
>
> - We compute the estimated FPR with the full validation data, as usual in OOD
> - We apply the correction
> - We compute the TPR using OOD validation data (as usual in OOD) but with a threshold that corresponds to the corrected FPR instead of the non-corrected FPR.
>
> We added a remark in the main manuscript in Section 4.4 (Remark 4.1) to clarify this point.
>
> ---
> # Question
>
> Classical CP scores rely on softmax (LAC, APS, RAPS), which can be related to prediction confidence. But it is also used as an OOD score [1]. In fact, many OOD scores are derived from softmax (energy [2], entropy [3], GEN [4]). More generally, as seen in OpenOOD's paper benchmark results, predictive uncertainty methods work well for OOD detection. It does not necessarily mean that one single metric can accurately capture both types of uncertainties, but it suggests that there may be a link between those two and that one is a good practical proxy for the other.
>
> Still, OOD scores are not directly related to uncertainties, e.g., DKNN [6], which is, in fact, very competitive for OOD detection.
>
> ---
>
> [1] Dan Hendrycks and Kevin Gimpel. A Baseline for Detecting Misclassified and Out-of-distribution Examples in Neural Networks.
>
> [2] Weitang Liu, Xiaoyun Wang, John D. Owens, and Yixuan Li. Energy-based Out-of-distribution Detection
>
> [3] Jie Ren, Peter J. Liu, Emily Fertig, Jasper Snoek, Ryan Poplin, Mark Depristo, Joshua Dillon, Balaji Lakshminarayanan. Likelihood Ratios for Out-of-Distribution Detection
>
> [4] Xixi Liu, Yaroslava Lochman, Christopher Zach. GEN: Pushing the Limits of Softmax-Based Out-of-Distribution Detection

---

> ### Comment · Reviewer_16UK · 2024-11-26
> **Thank the authors for answering my questions**
>
> Thank the authors for answering my questions and clarifying my doubts. I am happy to maintain my rating.

---

> > ### Author Response · Authors · 2024-11-28
> >
> > Thank you for taking the time to read our response; we are happy to have successfully clarified your doubts and comforted your initial judgment.

---

### Official Review · Reviewer_NLkM · 2024-11-06

**Soundness:** 3
**Presentation:** 2
**Contribution:** 1
**Rating:** 3
**Confidence:** 4

**Summary:**

This paper investigates the relationship between Out-of-Distribution (OOD) detection and Conformal Prediction (CP). They apply conformal prediction methods from prior work by Bates et al. (2022) to the task of Out-of-Distribution (OOD) detection, which is to distinguish in-distribution (ID) data from OOD data. They introduce metrics conformal AUROC and conformal FPR@TPR95 which provide conservative probabilistic guarantees in OOD detection tasks, particularly relevant for safety-critical applications. The authors then empirically validate the performance of these metrics on common OOD benchmarks.

**Strengths:**

The paper presents an application by adapting the conformal prediction methods from Bates et al. (2022) to OOD detection.

**Weaknesses:**

(1) The methodology largely follows the framework established by Bates et al. (2022) in the conformal prediction literature, with limited novel contributions specific to OOD detection. The primary innovation appears to be an application of existing CP methods to OOD tasks rather than a new approach.

(2) While the idea of combining OOD detection with CP is interesting, the approach itself and the resulting metrics are relatively straightforward extensions and may not represent a significant advancement in either OOD or CP methodologies.

minors:
(1) Line 135: In Equation (2), n_val is not defined and weird, should be probability. Similarly, n_val is used as probability in Line 296, Equation (10). Other places, it is used as the number of data points.

(2) A lot of places, for example, section 4.3.1, tau and t are used in mix to refer to the same thing.

**Questions:**

Given the significant overlap with Bates et al. (2022) 's methods, what specific contributions does this paper make beyond adapting CP to OOD tasks?

---

> ### Author Response · Authors · 2024-11-22
>
> We would like to thank reviewer NLkM for his or her constructive feedback on the writing and the technical details of the paper. We first addressed the main concerns, which is the overlap between our work and the work of Bates et al. (2022) (seen in the weaknesses and the question section). Then, we address minor comments.
>
> ---
> # Main concern
>
> ## The contributions are not limited to the corrected metrics
>
> This criticism is supported by the observation that the technical foundations of our corrected metrics rely on Bates et al. (2022). This observation is true, and we acknowledge it several times in the manuscript.  However, deriving corrected metrics is far from being the only contribution of this work. In addition to this contribution:
>
> - We point out that the common practice of OOD and Anomaly detection enables the use of this theoretical work. Drawing this link is a contribution.
> - We apply it to extensive OOD and Anomaly Detection benchmarks. You point out that the OOD baselines perform similarly to the vanilla metric. **While you consider it as a weakness, we argue that it is a strength**: it means that using corrected metrics, that provide **safety guarantees, comes with almost no cost** and could hence be used seamlessly in OOD. The conclusion is not the same for Anomaly Detection, which shows that current benchmarks are brittle when it comes to safety assessment, which is another strong insight.
> - We also use OOD scores to build new prediction intervals, which is **not related to the work of Bates et al. (2022)**. It opens a whole avenue for nonconformity score crafting, which is central to CP performances.
> - To us, perhaps the most important contribution is the **message** carried by the paper, supported by our several contributions : OOD and CP are closely related and should be used jointly. We believe that this message could have a significant impact on both fields, which are very dynamic and active research fields in AI.
>
> We agree that each contribution taken independently would not be a sufficient contribution for a whole paper. But they form a coherent whole that advocates for further exploration of the link between CP and OOD, which is, in our opinion, a very important insight and could have a **broad impact on future research in both fields**.
>
> ---
>
> # Minor remarks
>
> - Thank you for pointing out the notation problem with $n_val$. We corrected it in the revised version of the manuscript.
> - About $\tau$ and $t$: $\tau$ is used for threshold on the scores, and $t$ for threshold on the p  value. $t \in [0,1]$ whereas $\tau \in \mathbb{R}$, so each of them has its own use.

---

> > ### Comment · Reviewer_NLkM · 2024-11-25
> >
> > Thank the authors for their detailed response. I appreciate their contributions in applying conformal prediction methods to out-of-distribution detection and anomaly detection problems.
> > However, I maintain the perspective that the methodological novelty of the paper is limited and may not meet the standards required for acceptance at ICLR.

---

> ### Author Response · Authors · 2024-11-28
>
> Thank you for taking the time to read our response.
>
> We regret that you deem our paper not worthy of publication due to the lack of methodological novelty, regardless of the potential impact of its message. **We would like to make sure that you correctly caught that our contributions were not only about applying conformal prediction to OOD and AD, as you mentioned in your last message but also doing the converse and using OOD for CP**.
>
> Lastly, we would like to quote ICLR's reviewer guidelines: "Submissions bring value to the ICLR community when they convincingly demonstrate new, relevant, impactful knowledge (incl., empirical, theoretical, for practitioners, etc).". We admit that the paper does not bring many significant methodological novelties (though it brings several). Still, we would be disappointed that our message could not be heard in ICLR, despite its broad impact, just because of this reason. In our opinion, a methodologically simple paper with more potential impact is as worthy of publication as a paper of high technicality but with restricted application. But of course, it is only our opinion, and we undoubtedly respect yours

---

### Official Review · Reviewer_41Qn · 2024-11-06

**Soundness:** 3
**Presentation:** 3
**Contribution:** 2
**Rating:** 5
**Confidence:** 3

**Summary:**

The paper examines the relationship between conformal prediction (CP) and out-of-distribution (OOD) detection. It approaches the OOD detection problem from a conformal prediction perspective and, inspired by CP, proposes several modified metrics for OOD evaluation, testing them on various OOD benchmarks. Additionally, the paper constructs prediction sets for classification tasks, drawing on techniques from OOD detection.

**Strengths:**

* The paper is well-structured, well-written, and easy to read.

* The paper introduces modified metrics for OOD evaluation that are more robust when considered from a hypothesis testing perspective.

* The OOD detection problem is significant for ML research, making this topic relevant to the ICLR community.

* The exploration of the link between OOD and CP is informative.

**Weaknesses:**

* The main insight of the paper is that FPR@β can be interpreted as a p-value within a specific statistical hypothesis testing framework. By building on the work of [1], the paper proposes a corrected estimator for this metric to improve robustness. While this approach is interesting, I believe it is insufficient to warrant publication. Firstly, much of the technical foundation relies on [1], and the corrected metric for OOD detection performs similarly to the classical version. Therefore, I do not find the current version sufficiently novel for publication.

* Computing the proposed metric for OOD detection requires access to extra validation sets.

Minor Comments:
Line 22: FPR? (not FRP)

[1]Stephen Bates, Emmanuel Candès, Lihua Lei, Yaniv Romano, and Matteo Sesia. Testing for Outliers
with Conformal p-values, May 2022. URL http://arxiv.org/abs/2104.08279.

**Questions:**

1. Is there a scenario where classical metrics like FPR@β or AUROC might fail for OOD evaluation, but your conformal metrics accurately capture the evaluation? How do the experimental results in Section 4 contribute to safer OOD evaluation?

2. When using OOD scores as nonconformity scores in CP, can you still obtain theoretical guarantees for the prediction sets?

---

> ### Author Response · Authors · 2024-11-22
>
> We would like to thank reviewer 41Qn for appreciating the writing of our paper and recognizing its importance and relevance to the ICLR community. We first answer your main concerns (also discussed in the general answer).
>
> ---
> # Main concerns
>
> ## 1. The contributions are not limited to the corrected metrics
>
> This criticism is supported by the observation that the technical foundations of our corrected metrics rely on Bates et al. (2022). This observation is true, and we acknowledge it several times in the manuscript.  However, deriving corrected metrics is far from being the only contribution of this work. In addition to this contribution:
>
> - We point out that the common practice of OOD and Anomaly detection enables the use of this theoretical work. Drawing this link is a contribution.
> - We apply it to extensive OOD and Anomaly Detection benchmarks. You point out that the OOD baselines perform similarly to the vanilla metric. **While you consider it as a weakness, we argue that it is a strength**: it means that using corrected metrics, that provide **safety guarantees, comes with almost no cost** and could hence be used seamlessly in OOD. The conclusion is not the same for Anomaly Detection, which shows that current benchmarks are brittle when it comes to safety assessment, which is another strong insight.
> - We also use OOD scores to build new prediction intervals, which is **not related to the work of Bates et al. (2022)**. It opens a whole avenue for nonconformity score crafting, which is central to CP performances.
> - To us, perhaps the most important contribution is the **message** carried by the paper, supported by our several contributions : OOD and CP are closely related and should be used jointly. We believe that this message could have a significant impact on both fields, which are very dynamic and active research fields in AI.
>
> We agree that each contribution taken independently would not be a sufficient contribution for a whole paper. But they form a coherent whole that advocates for further exploration of the link between CP and OOD, which is, in our opinion, a very important insight and could have a **broad impact on future research in both fields**.
>
> ## 2. We do not need extra validation (or calibration) data
>
> We should have insisted more on that point because it is another advantage of our method: **it does not require extra validation data**.
>
> Computing the conformal FPR only requires a correction on the estimated FPR that does not rely on a third calibration dataset. It is not like in CP where we need a calibration dataset to find a threshold based on nonconformity scores obtained on calibration data, that is subsequently used to provide CP confidence intervals. Here, there are no confidence prediction intervals; we only use CP theory to obtain probabilistic guarantees on the FPR. The steps are as follows:
>
> - We compute the estimated FPR with the full validation data, as usual in OOD
> - We apply the correction
> - We compute the TPR using OOD validation data (as usual in OOD) but with a threshold that corresponds to the corrected FPR instead of the non-corrected FPR.
>
> We added a remark in the main manuscript in Section 4.4 (Remark 4.1) to clarify this point.
>
> ---
>
> # Questions
>
> 1. In Figure 2, we show that the AUROC is overestimated using classical evaluation of CIFAR10 vs SVHN. The experimental results of Section 4 contribute to safer benchmarks because they limit the risk of the AUROC and FPR being overestimated, which is highly undesirable for safety-critical applications. Certification processes that ensure safety levels for any safety critical systems seek guarantees that the estimations are conservative with controlled probabilities. This is not provided by classical metrics.
> 2. The guarantees stemming from conformal prediction are automatically preserved in whatever score we use, even if it is an OOD score. This is one of the strengths of the approach: we highlight that every new OOD score is a potential nonconformity score for CP.

---

> > ### Comment · Reviewer_41Qn · 2024-11-24
> > **Thanks for your rebuttal**
> >
> > Dear Authors,
> >
> > Thank you for your response. While I appreciate the informative message of your work and have decided to increase my score, I still believe the technical contribution is not sufficient for a new publication.
> >
> > Best,
> > Reviewer

---

> ### Author Response · Authors · 2024-11-28
>
> Thank you very much for increasing your score, though not up to acceptance.
> We regret that you deem our paper not worthy of publication because you judge the technical contribution as not sufficient, regardless of the potential impact of its message.
>
> Lastly, we would like to quote ICLR's reviewer guidelines: "Submissions bring value to the ICLR community when they convincingly demonstrate new, relevant, impactful knowledge (incl., empirical, theoretical, for practitioners, etc).". We admit that our work is not highly technical (though it required significant empirical and implementation efforts to run all the benchmarks and to adapt OOD scores to CP, respectively). Still, we would be disappointed that our message could not be heard in ICLR, despite its broad impact, just because of this reason. In our opinion, a technically simple paper with more potential impact is as worthy of publication as a paper of high technicality but with restricted application. But of course, it is only our opinion, and we undoubtedly respect yours.

---

### Author Response · Authors · 2024-11-22
**General response to reviewers**

We would like to thank the reviewers for their constructive feedback. We took into account all the minor remarks and updated a corrected revision of the manuscript. Thank you for carefully reading the manuscript and quoting these typos!

Two concerns are shared by three of the four reviewers:

1. They deem that the contribution is limited due to the overlap with Bates et al. (2022) for corrected metrics
2. They are concerned with the need for extra-validation data to compute the proposed metrics.

## 1. The contributions are not limited to the corrected metrics (reviewers 41Qn, NLkM and 16UK)

This criticism is supported by the observation that the technical foundations of our corrected metrics rely on Bates et al. (2022). This observation is true, and we acknowledge it several times in the manuscript.  However, deriving corrected metrics is far from being the only contribution of this work. In addition to this contribution:

- We point out that the common practice of OOD and Anomaly detection enables the use of this theoretical work. Drawing this link is a contribution.
- We apply it to extensive OOD and Anomaly Detection benchmarks. Some reviewers point out that the OOD baselines perform similarly to the vanilla metric. **While they consider it as a weakness, we argue that it is a strength**: it means that using corrected metrics, that provide **safety guarantees, comes with almost no cost** and could hence be used seamlessly in OOD. The conclusion is not the same for Anomaly Detection, which shows that current benchmarks are brittle when it comes to safety assessment, which is another strong insight.
- We also use OOD scores to build new prediction intervals, which is **not related to the work of Bates et al. (2022)**. It opens a whole avenue for nonconformity score crafting, which is central to CP performances.
- To us, perhaps the most important contribution is the **message** carried by the paper, supported by our several contributions : OOD and CP are closely related and should be used jointly. We believe that this message could have a significant impact on both fields, which are very dynamic and active research fields in AI.

We agree that each contribution taken independently would not be a sufficient contribution for a whole paper. But they form a coherent whole that advocates for further exploration of the link between CP and OOD, which is, in our opinion, a very important insight and could have a **broad impact on future research in both fields**.

## 2. We do not need extra validation (or calibration) data (reviewers 41Qn and NLkM)

We should have insisted more on that point because it is another advantage of our method: **it does not require extra validation data**.

Computing the conformal FPR only requires a correction on the estimated FPR that does not rely on a third calibration dataset. It is not like in CP where we need a calibration dataset to find a threshold based on nonconformity scores obtained on calibration data, that is subsequently used to provide CP confidence intervals. Here, there are no confidence prediction intervals; we only use CP theory to obtain probabilistic guarantees on the FPR. The steps are as follows:

- We compute the estimated FPR with the full validation data, as usual in OOD
- We apply the correction
- We compute the TPR using OOD validation data (as usual in OOD) but with a threshold that corresponds to the corrected FPR instead of the non-corrected FPR.

We added a remark in the main manuscript in Section 4.4 (Remark 4.1) to clarify this point.

When relevant, we copy these discussions in dedicated answers to facilitate the reading of our response, where we also address the reviewer's specific concerns.

---

### Meta-Review · Area_Chair_MiJt · 2024-12-20

**Metareview:**

This paper studies the relationship between conformal prediction (CP) and out-of-distribution (OOD) detection. It proposes to modify some metrics for OOD detection by taking a CP perspective. These were evaluated  It also investigates how to the OOD prediction scores can be used to improve prediction sets constructed by CP for classification tasks.

Strengths:
- paper well written,
- it proposes more robust metrics for OOD evaluation,
- it makes links with OOD and conformal prediction (CP).

Weaknesses:
- innovation limited due to the strong connection with Bates et al, 2022,
- need of extra-validation sets,
- the paper straightforward extensions when linking OOD and CP.

The authors have provided additional precisions during the rebuttal, in particular they raised 2 main points: (i) theirs contributions are not limited to the corrected metrics (in particular they mention that they go beyond Bates et al. and provide improvements for understandings the links between OOD and CP), (ii) they do not need extra-validations datasets.

During discussion, reviewers have appreciated the idea of studying the links between OOD detection and CP. However, a majority analysis for saying that the paper presents in its current form a limited contribution in the sense that the connections between CP and OOD are not sufficiently studied and more technical contributions, deeper analysis and extended evaluations are suitable for acceptance to ICLR. The last experiments did not convince totally.

The remaining negative weaknesses focus on the core contributions of the paper which appear to have limitations requiring further improvements, I propose then rejection.
However, the idea of studying the connections between OOD detection and CP seems very promising, I encourage the authors to improve their work for other venues.

**Additional Comments On Reviewer Discussion:**

During discussion, Reviewer u4qk confirmed that he was satisfied with the answers of the authors and kept his score at 6.

Reviewer 41Qn increased his score to 5 estimated that the contribution is not sufficient, he mentioned during the discussion that he thinks that the paper has potential and room for improvement and that the authors should strengthen their message with more technical contributions to reinforce the paper.

Reviewer NLkM, who was the most negative (rating 3), estimates that the connections between OOD detection and CP are not sufficiently analyzed, either because the contribution are too close to (Bates et al. 2022)'s work or begin a straightforward use of CP, or the use of OOD scores for conformal prediction deserves more theoretical analysis, the novel experiments provided were not sufficient.

Reviewer 16UK was positive but identified some limitations but maintained his rating of 6, but did not support the paper during discussion.

Overall, the weaknesses raises focused on the core contribution of the paper which outweighs in my opinion the merits, rejection is then proposed.

---

### Decision · Program_Chairs · 2025-01-22

Reject